# Using the antibody-antigen binding interface to train image-based deep neural networks for antibody-epitope classification

**Daniel R. Ripoll**[1,2], **Sidhartha Chaudhury**[1,3], **Anders Wallqvist**[1]*

1 DoD Biotechnology High Performance Computing Software Applications Institute, Telemedicine and Advanced Technology Research Center, U.S. Army Medical Research and Development Command, Fort Detrick, Maryland, United States of America, 2 Henry M. Jackson Foundation for the Advancement of Military Medicine, Inc. (HJF), Bethesda, Maryland, United States of America, 3 Center for Enabling Capabilities, Walter Reed Army Institute of Research, Silver Spring, Maryland, United States of America

* sven.a.wallqvist.civ@mail.mil

**Data Availability Statement:** All relevant data are within the manuscript and its Supporting Information files. Software and examples are

## Abstract

High-throughput B-cell sequencing has opened up new avenues for investigating complex mechanisms underlying our adaptive immune response. These technological advances drive data generation and the need to mine and analyze the information contained in these large datasets, in particular the identification of therapeutic antibodies (Abs) or those associated with disease exposure and protection. Here, we describe our efforts to use artificial intelligence (AI)-based image-analyses for prospective classification of Abs based solely on sequence information. We hypothesized that Abs recognizing the same part of an antigen share a limited set of features at the binding interface, and that the binding site regions of these Abs share share common structure and physicochemical property patterns that can serve as a "fingerprint" to recognize uncharacterized Abs. We combined large-scale sequence-based protein-structure predictions to generate ensembles of 3-D Ab models, reduced the Ab binding interface to a 2-D image (fingerprint), used pre-trained convolutional neural networks to extract features, and trained deep neural networks (DNNs) to classify Abs. We evaluated this approach using Ab sequences derived from human HIV and Ebola viral infections to differentiate between two Abs, Abs belonging to specific B-cell family lineages, and Abs with different epitope preferences. In addition, we explored a different type of DNN method to detect one class of Abs from a larger pool of Abs. Testing on Ab sets that had been kept aside during model training, we achieved average prediction accuracies ranging from 71–96% depending on the complexity of the classification task. The high level of accuracies reached during these classification tests suggests that the DNN models were able to learn a series of structural patterns shared by Abs belonging to the same class. The developed methodology provides a means to apply AI-based image recognition techniques to analyze high-throughput B-cell sequencing datasets (repertoires) for Ab classification.

available through a Github repository https://github.com/dripoll53/AbsFngP/.

**Funding:** Support for this research was provided by the US Army Medical Research and Development Command (mrdc.amedd.army.mil/) under Contract No.W81XWH20C0031 (AW), and the U.S. Department of Defense (DoD) High Performance Computing Modernization Program (www.hpc.mil/) (AW, DRR). The funders had no role in study design, data collection and analysis, decision to publish, or preparation of the manuscript.

**Competing interests:** The authors have declared that no competing interests exist.

## Author summary

The ability to take advantage of the rapid progress in AI for biological and medical application oftentimes requires looking at the problem from a non-traditional point-of-view. The adaptive immune system plays a key role in providing long-term immunity against pathogens. The repertoire of circulating B-cells that produce unique pathogen-specific antibodies in an individual contains immense information on both the status of the immune response at particular time and that individual's immune history. With high-throughput sequencing, we can now obtain Ab sequences for thousands of B cells from a single patient blood sample, but functionally characterizing antibodies on this scale remains on daunting task. Here, we propose to use AI to functionally classify Abs from sequence alone by re-casting this classification problem as an image recognition problem. Just as traditional image recognition involves training AI to distinguish different types of objects, we sought to use AI to distinguish different types of Ab-antigen binding interfaces. Towards that end, we generated ensembles of Ab structures from sequence, and generated 2-D 'fingerprints' of each structure that captures the essential molecular and chemical structure of the Ab binding site regions, and trained a Convolution and Deep Neural Network based AI model to classify Ab fingerprints associated with different functional characteristics. We applied this DNN-based approach to accurately predict antibody family lineage and epitope specificity against Ebola and HIV-1 viruses, and to detect sequence-diverse antibodies with similar binding properties as the ones we used for training.

This is a *PLOS Computational Biology* Methods paper.

## Introduction

The human body contains approximately 3 liters of serum. If we consider that the normal levels of IgG in human serum range from 7 to 15 g/L [1], and that the molecular weight of an IgG molecule is ~1.5E5 g/mol, we can estimate that the human immune system is capable of producing on the order of $10^{20}$ antibodies (Abs) in response to a viral infection. Only a small fraction of these Abs bind strongly to any given antigen, and an even smaller fraction is capable of neutralizing an infection. If we could rapidly screen and identify Abs with desirable properties from an individual's entire set of Abs (i.e., an Ab repertoire), we would be able to accelerate and improve the development of vaccines, therapeutics, and assays. During the past few years, high-throughput sequencing of B-cell immunoglobulin repertoires has emerged as a valuable tool in studying the evolution of Abs upon infection, accelerating the process of antigen-specific monoclonal Ab (mAb) discovery [2,3], and developing disease diagnostics [4,5].

A main objective of our research is to assess if immune-response properties of Abs can be inferred from high-throughput sequencing data of B-cell repertoires using computational tools. The sequence carry the information needed to describe the Ab binding site, both in terms of physicochemical properties of amino acid residues (e.g., charge and aromaticity) and their structural arrangements.

We hypothesize that the complement of residues in the Ab-antigen binding interface determines the preference for a particular epitope on the antigen, and that Abs that have evolved

from unrelated clonotypes to recognize this epitope, are likely to share common structural patterns and physiochemical properties even if they differ in their binding modes.

How can we test this hypothesis? Experimental determination of the 3-D structure of every Ab in a repertoire is unfeasible. Recent advances in computational algorithms have allowed researchers to produce structural models for thousands of Abs to complete repertoires [6,7]. Analyses of these models has led to a better understanding of the structural profiles of naïve B-cells, and the structural changes of the complementarity-determining regions (CDRs) that reshape the Ab binding sites during B-cell differentiation. However, even if we obtain a computational or experimental 3D model of the Abs, it is non-trivial to infer if two clonally distinct Abs can recognize the same or overlapping epitope(s) using different binding modes, since this task cannot be resolved using sequence based approaches. To partly overcome this problem, we can produce large conformational ensembles of the Abs employing molecular modeling techniques to account in part for uncertainties associated with the flexibility of the Ab CDRs. Then, by comparing multiple conformers of the Abs we can assess the likelihood that key residues important for antigen recognition adopt similar conformations. However, comparing large sets of conformers in search for common structural features still represents a daunting task.

The data derived from immunological studies typically represent the result of ongoing stochastic and multifactorial processes that is often difficult to decipher. Artificial Intelligence (AI) methods are potentially well suited to address these types of problems [8]. Thus, machine learning (ML) approaches have already been used to analyze and classify information derived from cells expressing adaptive immune receptors [9]. Some of these ML applications include predicting peptide presentation by T cells [10–13], affinity of peptide binding to Major Histocompatibility Complex molecules [14,15], and binding affinity of neutralizing Abs [16]. Additionally, deep learning techniques have also been used in Ab paratope prediction [17].

In this work, we describe how transforming a search for structural similarities among Ab conformers into a search for common patterns among multiple images of the Ab binding site region makes the computational problem addressable using machine learning methods (S1 Fig). Importantly, this transformation allow us to take advantage of powerful AI methods that are being developed for image recognition such as DNNs.

Our approach consists of reducing the complexity associated with a full comparison of large sets of Ab 3-D models by introducing a simplified representation of an Ab binding site, termed a *fingerprint*. To generate a fingerprint, we focus on the Ab residues that delineate the Ab-antigen binding site. The predicted 3-D structural arrangement of these Ab residues are projected onto a 2-D plane that intersects the Ab-antigen binding interface, and then colored based on properties that are important for binding, e.g., physicochemical property such as residue charge, hydrophobicity, and aromaticity as well as the distance from the binding interface. The fingerprints are derived from 3-D models representing the unbound conformation of an Ab. No attempt was made to account for excluded volume effects on the CDR loops due to the presence of antigens. The constructed fingerprint creates a 2-D image of the Ab-antigen binding interface that can be saved for subsequent analysis. The collection of fingerprints generated from an ensemble of predicted structures is hypothesized to capture the conformational flexibility associated with the Ab CDRs, a key feature required for epitope recognition and binding. Given that only a few residues in the Ab-antigen interface make a major contribution to the binding energy [18], it is expected that the number of residue motifs that Abs can employ for recognition and binding to the same epitope is limited. Thus, the fingerprints from the group of Abs that bind to this epitope are expected to share a reduced set of key features. Under this assumption, we can customize the existing framework of Deep Neural Network (DNN) image

analysis [19], deploy it to detect similarities among fingerprints, and carry out classifications of Abs.

## Results

### Overview of the methodology

Based on the assumption that residues forming the Ab binding site region underlie the key elements that drive Ab-antigen complex formation, we assessed if Abs with similar immunological characteristics share common structural and physicochemical features at the Ab-antigen binding interface. Specifically, as shown schematically in S1B Fig, we evaluated in a stepwise fashion, progressing from less to more complex problems, the needed characteristics of an AI methodology that would ultimately be required for assessing high-throughput B cell immune-sequencing data. Specifically, we aimed to investigate if the developed approach could address the following of four problems without using explicitly sequence information or residue connectivity: 1) Can we identify Abs based solely on a reduced number of features from Ab binding site region? 2) Can we identify the family lineage of Abs using features from the Ab binding site regions? *3)* Can we detect Abs with common function, such as sharing similar binding preferences or specificity to an antigen? *4)* Having a set of Abs with a common desired property, can we search a database of sequences and detect Abs that share this property based solely on fingerprints?

For our analyses, we collected data from a series of studies on B cell repertoires associated with viral infections. We focused primarily on Ebola virus (EBOV), and human immunodeficiency virus (HIV) studies for which retrievable paired antibody heavy- and light-chain sequencing data are available in public databases, together with data from binding affinity, neutralization assays, and additional structural studies on Ab-antigen complexes involving these Abs that has already been reported.

To carry out the computational experiments, we followed a series of steps delineated in the schematic diagram presented in Fig 1. Briefly, we generated thousands of 3-D models for each Ab to construct the corresponding fingerprint of the binding site region. Then, we used these fingerprints to train DNN models to classify the antibodies according to certain characteristics, such as their preference for a given epitope, or the particular naïve B cell from which the B cell that encoded the Ab originated, i.e., the family lineage [20]. The detailed descriptions of all steps in this process are provided in the Materials and Methods.

We designed a series of computational experiments with varying degree of complexity to evaluate the feasibility and accuracy of using this approach to assess the methodological questions above. First, we evaluated if trained DNN models could associate correctly fingerprint images from two Abs. As a second test, we evaluated if trained DNN model could detect the family lineage of Abs based on fingerprint features. Third, we evaluated the performance of trained DNNs as a tool for classification of antibodies effective against Ebola (anti-EBOV Abs) based on their binding specificity for certain epitopes, using three different scenarios. Similarly, we carried out a similar type of study by training DNNs to classify anti-HIV GP120 Abs based on the epitope they recognize. Finally, we tested the predictive capability of one-class classification DNNs by training them to recognize a single class of Abs, and then, used that DNN model to infer what Abs from a larger independent testing set belong to the learned class. It is worth noting that, with exception of the first experiment, assessment of the trained DNN models in all other experiments were produced using conditions that are more stringent. We grouped Abs based on a specified property into the categories or classes that the DNN were supposed to learn. Then, we used the fingerprint images from a subset of Abs belonging each class for training and validation of the DNNs. Finally, we used the fingerprints from the

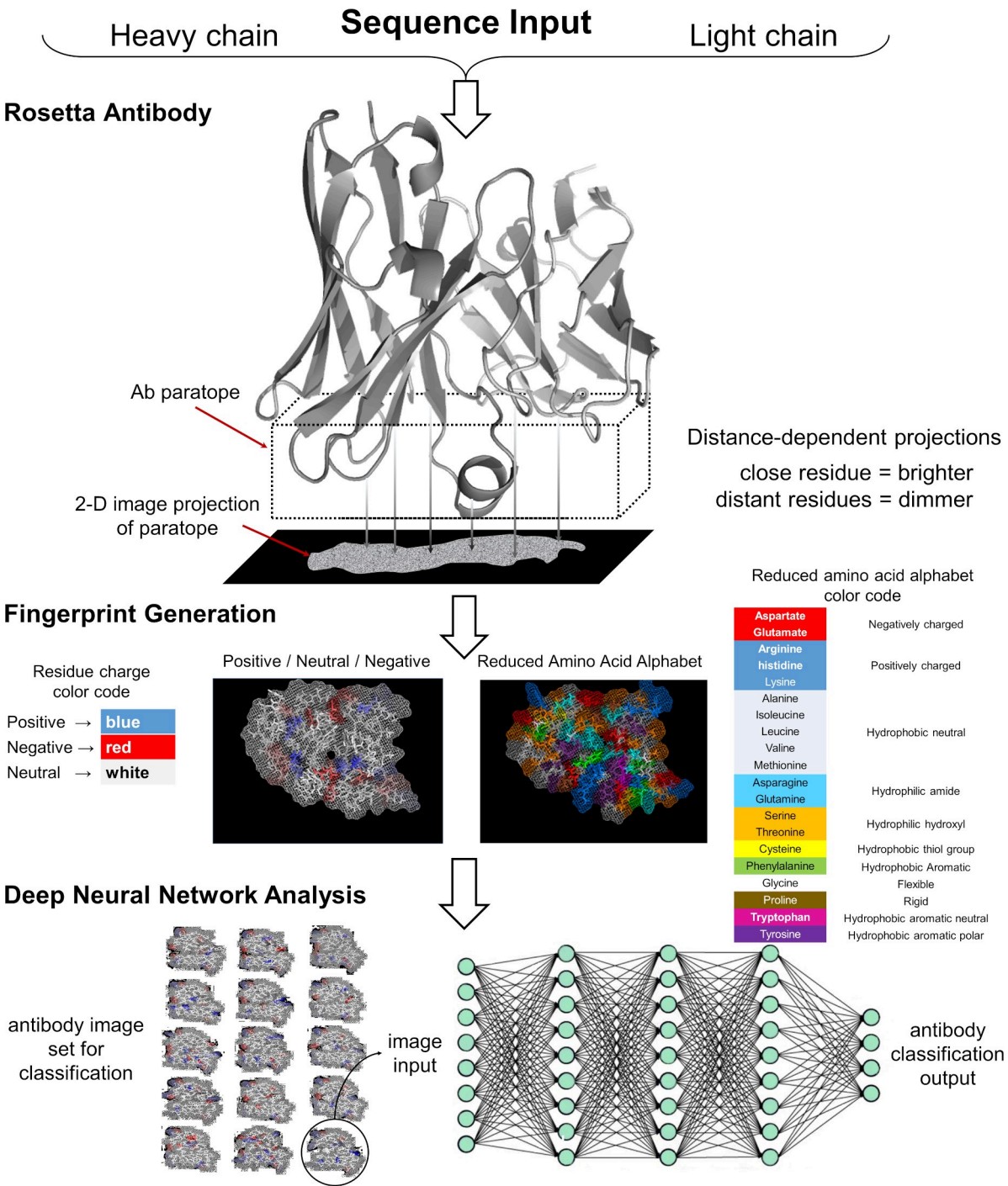

**Fig 1. Schematic overview describing the steps required to generate fingerprints for Deep Neural Network image analysis.** We used the Rosetta Antibody software to generate multiple 3-D models of a particular Ab or one of its antibody binding fragment (FAB) using the light and heavy chain sequences as input data. For each 3-D model, we used PYMOL to produce a fine grid perpendicular to the main axis of the Ab, which intersects the Ab binding site region. We selected amino acid residues from the model that lies within a distance of 20 Å from the grid, and their atoms were projected onto the 2-D grid and displayed using a "dot" representation. The image was then colored according to the desired color-scheme using either a charge or an amino acid property based representation. The resulting image was then stored as an image file. The transformation of the sequence into an image allowed us to train DNNs models for Ab classification purposes using collections of fingerprint sets from multiple Abs.

**Table 1. Summary of examined Tasks, the number of Abs, DNN models, and fingerprint images used in each Task.**

| Task | N Abs | Number of DNN models | Number of images per model | | | Accuracy [Testing] (SD) | Other |
|---|---|---|---|---|---|---|---|
| | | | Training (SD) | Validation (SD) | Testing (SD) | | |
| *Classification Tasks: Differentiating Ab properties* | | | | | | | |
| Distinguish two Abs | 2 | 10 | 395 (22) | 93 (15) | 121 (9) | 0.96 (0.05) | |
| Prediction of Ab lineage | 28 | 132 | 2170 (90) | 507 (89) | 1000 (0) | 0.71 (0.12) | |
| Differentiating Abs that recognize two different epitopes in EBOV GP | 29 | 60 | 950 (63) | 276 (31) | 490 (80) | 0.84 (0.12) | |
| Differentiating Abs that recognize three different epitopes in EBOV GP | 30 | 20 | 1522 (126) | 449 (49) | 769 (102) | 0.71 (0.09) | |
| Differentiating Abs that recognize two different epitopes in HIV GP120/GP41 | 28 | 50 | 546 (63) | 173 (25) | 386 (89) | 0.86 (0.10) | |
| *One-Class Classification Tasks: Detecting specific Abs* | | | | | | | |
| Detection of Abs from a specific lineage from a collection of Abs from 9 other lineages | 25 | 12 | 663 (152) | 5184 (962) | 1193 (512) | 0.80 (0.27) | |
| Detection of Abs from a specific lineage from a large collection of different Abs | >200 | 13 | 696 (137) | 12907 (1823) | 2265 (229) | 0.95 (0.07) | |
| Detection of Abs from the same competition group but from different lineages | >200 | 100 | 233 (0) | 12656 (301) | 5412 (309) | N/A | 4 Ab detected |

remaining Abs that were not used for training purposes to assess the predicted capabilities of the trained DNN models. Table 1 summarizes the computational experiments described in this work, detailing for each of them the number of DNN models produced, the number of antibodies used, and the average number of fingerprints allocated to the training, validation, and testing sets, and their standard deviations.

## Differentiating two antibodies based on their fingerprints

Initially, we built and tested a DNN algorithm to identify fingerprint images from two Abs using the Keras application program interface [21]. We implemented a Python script that was built upon a pre-trained residual neural network architecture ResNet-50 [22] designed for image classification. Details on the procedure implemented to train the DNNs are given in the Materials and Methods Section: *Training and testing DNN for antibody classification based on fingerprints*.

We first carried out a series of computer experiments to assess the ability of the fingerprints to differentiate pairs of well-characterized EBOV Abs. Table 2 provides a summary of the ten different computer experiments developing DNN models to distinguish between two Abs. For training of these models, we used fingerprints colored according to the charge-residue code. The fingerprints from each Ab were split into three datasets: training, validation, and testing, indicated as $N_{training}$, $M_{validation}$, and $N_{test}$, respectively, in Table 2. In addition, we used an enhancement technique to increase the number of images in the training and validation datasets. In the initial training and validation phase, we selected the number of learning cycles or epochs to be around 30. Fig 2 shows the rapid improvement and convergence of the loss function versus the number of epochs for the DNN models in Table 2. Note that we set aside a separate testing set that was not used during training and validation calculations to reduce the risk of overfitting and to provide a more stringent evaluation of the developed DNN models.

For the pair-wise differentiation of Abs in Table 2, we found that the predictions correctly identify the images of each Ab with an average accuracy of 0.96 (standard deviation [SD] 0.05), and a computed Cohen's Kappa coefficient, κ, of 0.94 (SD 0.09). The use of the image

**Table 2. Summary of 10 DNN models trained to differentiate between fingerprints belonging to pairs of antibodies.** Fingerprints were generated using the charge coloring scheme shown in Fig 1.

| DNN Model | EBOV antibody pair[a] | | Training and Validation Image Set Sizes[b] | | | | Training Results | Testing Results | | |
|---|---|---|---|---|---|---|---|---|---|---|
| | $Ab_1$ | $Ab_2$ | $N_{training}$ | $M_{validation}$ | Enhanced Image Sets | | $A_{val}$[c] | $N_{test}$[d] | $A_{test}$[e] | κ |
| | | | | | $eN_{training}$ | $eM_{validation}$ | | | | |
| 1 | 15791 | 15848 | 446 | 114 | 50000 | 10000 | 1.00 | 140 | 1.00 | 1.00 |
| 2 | 15916 | 15964 | 406 | 74 | 50000 | 5000 | 1.00 | 120 | 0.94 | 0.92 |
| 3 | 15780 | 15978 | 382 | 98 | 37800 | 6000 | 0.92 | 120 | 0.84 | 0.74 |
| 4 | 16028 | 15954 | 380 | 112 | 37800 | 6000 | 1.00 | 108 | 0.94 | 0.92 |
| 5 | 15758 | 16042 | 366 | 102 | 40000 | 10000 | 1.00 | 132 | 0.89 | 0.84 |
| 6 | 15912 | 15951 | 390 | 90 | 37800 | 6000 | 0.99 | 120 | 0.97 | 0.96 |
| 7 | 16042 | 15978 | 387 | 99 | 40000 | 5000 | 1.00 | 114 | 1.00 | 1.00 |
| 8 | 15956 | 15791 | 407 | 73 | 50000 | 5000 | 1.00 | 120 | 1.00 | 1.00 |
| 9 | 15785 | 15966 | 406 | 74 | 50000 | 5000 | 1.00 | 120 | 1.00 | 1.00 |
| 10 | 15935 | 16038 | 382 | 98 | 37800 | 6000 | 1.00 | 120 | 1.00 | 1.00 |
| Average (SD) | | | 395 (22) | 93 (15) | 43120 (5982) | 6400 (1955) | 0.99 (0.03) | 121 (9) | 0.96 (0.05) | 0.94 (0.09) |

[a] Numbers under columns $Ab_1$ and $Ab_2$ correspond to the antibody identification labels (ADI) used by Bornholdt *et al.* [23].

[b] $N_{training}$ and $M_{validation}$ indicate the number of distinct fingerprints selected for training and validation; $eN_{training}$, and $eM_{validation}$ indicate the augmented number of images used for training and validation.

[c] $A_{val}$; best accuracy obtained by the DNN model on the validation set.

[d] $N_{test}$; number of images in the testing set. In this particular experiment, the fingerprint images belong to the pair of Abs listed, but were not included in the training or validation.

[e] $A_{test}$; accuracy obtained by the DNN model on the test set.

enhancement technique improved the results compared to calculations where the same training, validations and testing sets were used, but without image enhancement; for these latter calculations we found an accuracy of 0.87 (SD 0.14), and κ of 0.81 (SD 0.26) on the test image sets. In summary, DNNs were able to discriminate the Abs based on their fingerprints with a

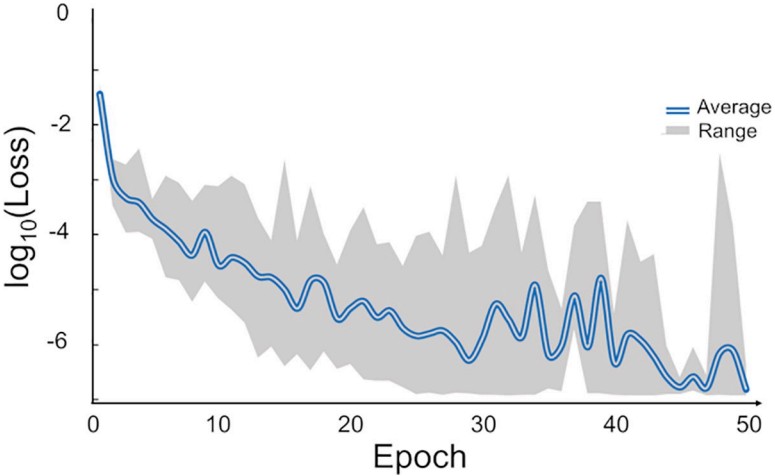

**Fig 2. Variation of the loss function for DNN models with the number of learning cycles.** The compound blue line represents average loss per epochs during training of 10 DNN models. Top and bottom of the gray area correspond to the maximum and minimum limits of the loss at each epoch for ten models. After about 30 epochs, there was no improvement in the loss function and we typically terminated the training at 30 epochs.

high accuracy. The results also indicate that the DNNs models are able to "learn" or identify alternative conformations from an initial set of images of the complementarity-determining regions (CDRs), and extrapolate this knowledge to identify new conformers not seen during training.

As noted earlier, all the predictions we report through the rest of this study were produced using testing conditions that are more stringent. Thus, in the following, our test sets contain fingerprint images from Abs corresponding to the same classes of Abs that the DNN is supposed to learn, but none of fingerprints selected for testing correspond to Abs included in either the training or validation sets.

To generate these sets we followed the protocol outlined in Fig 3 and described in the Materials and Method Section: *Training and testing DNN for antibody classification on fingerprints*.

## DNN prediction of family lineage from fingerprints of anti-EBOV Abs

In our second set of experiments, we investigated if we could train a DNN model to infer the family lineage of an Ab based solely on fingerprint features. Here, we expect the DNN to learn features of the binding site regions of various Abs from B cell families, where each family has a different common progenitor B cell. Then, we ask the DNN model to classify new Abs into one of the known families. We note that the detection of family lineages is easily achieved with computational tools based on sequence analysis. Our objective, however, was to determine the ability of DNNs to learn to associate members of the lineage family using similarities in the image patterns based on the arrangement of color on the fingerprints.

For the present study, we selected a small set of anti-EBOV Abs obtained from a survivor of the 2014-Zaire outbreak [23]. For these Abs, we detected ten lineage families as calculated with BRILIA [24] and described in the Materials & Methods Section: *Set of anti-EBOV antibodies*. Fig 4 illustrates the evolution from a single germline sequence and the fingerprint of four Abs from a single-family lineage. Fig 4A shows amino acid substitutions in the heavy-chain CDR3 loops of the Abs related to the germline gene. In Fig 4B, we display a small sample of fingerprints obtained from three 3-D models of each Ab from this family to provide a graphic example of the differences and similarities that exist within a family. Our goal was to train a DNN

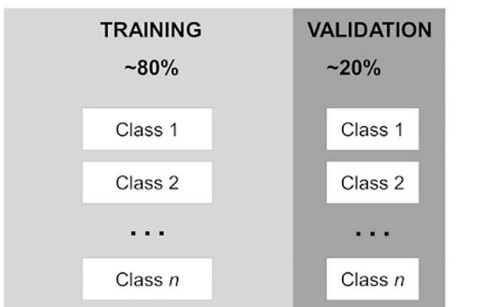
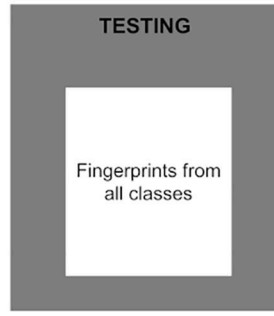

**Fig 3. Schematic diagram of the allocation of fingerprints into training, validation, and testing sets.** *Antibody assignment*: Antibodies are randomly split into two fractions: training/validation and testing. *Fingerprints assignment*: The fingerprint images of an Ab selected for testing are added to a common pool in the test set. If the Ab was selected for training/validation, its fingerprints are divided into two fractions: each fraction is added to specific pools in the training and the validation sets associated with the Ab class.

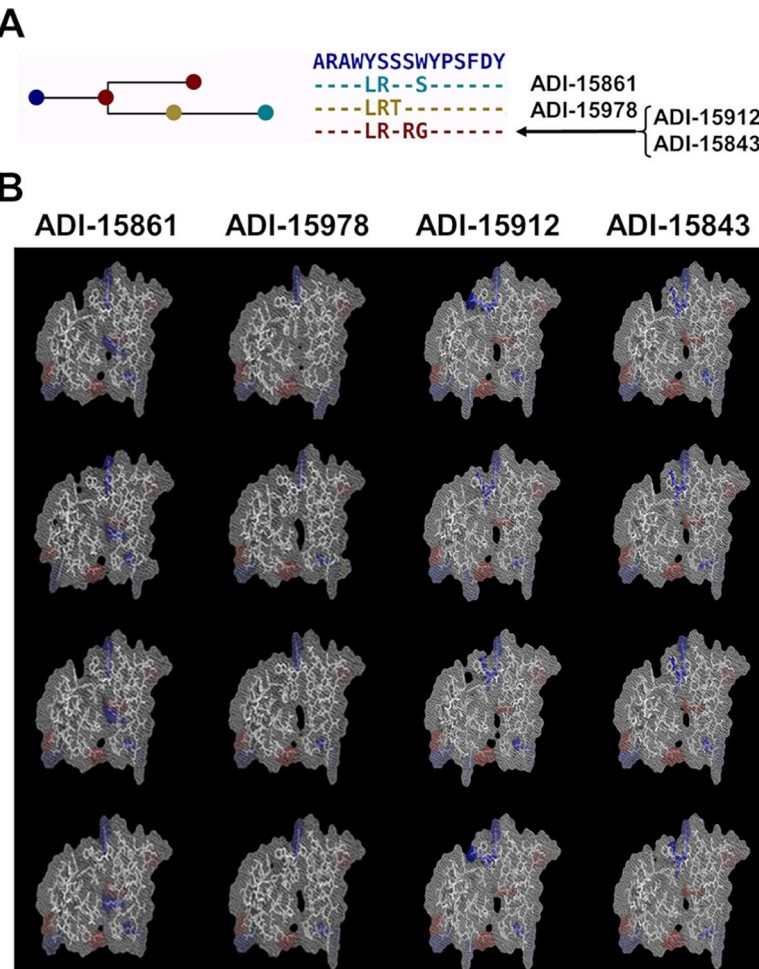

**Fig 4. Set of four antibodies associated with one family lineage.** (A) The graph highlights the amino acid substitutions in the heavy chain CDR3 region of the Abs with respect to the germline gene. Abs ADI-15912 and ADI-15843 share the same CDR3 sequence. (B) Each column shows three fingerprints for each Ab of the family showing how the amino acid substitutions listed in (A), and conformational changes in the models affect the fingerprints.

model to associate subsets of Abs fingerprints with their corresponding family lineage, and then, use the trained DNN model to make predictions based on not previously seen fingerprints from other Abs belonging to these lineages.

To produce the DNN models for detection of Ab family lineages, we chose 28 Abs (see S1 Table) belonging to ten families of anti-EBOV antibodies, together with their fingerprints (S2 Fig) to carry out training of DNNs, followed by testing of the predictive capabilities of the optimized models for family assignment. Table 3 provides a statistical summary of 80 DNN models, where each of the models was trained and tested using different datasets of fingerprints colored according to the charge coloring code. We used all the DNN models to produce 120,000 predictions of which 85,484 were correct, leading to a global accuracy ($A_{test}$) of 0.71 (SD 0.12).

In order to characterize any dependence on the DNN model prediction accuracy on the lineage family itself, we carried out additional statistical analysis. Table 4 lists separate statistical data that provide a quantitative evaluation of the performance of the models in a multiclass classification. Local measures of precision, recall, and F1-scores relate to the predictive

**Table 3. Training of DNNs for recognition of ten lineages.** Statistical summary for 80 DNN models used for classification of 28 antibodies belonging to ten family lineages using fingerprints colored according to the charge coloring code.

| DNN Model Training | Testing Results | | | |
|---|---|---|---|---|
| $\langle A_{val}\rangle^a$ | $\langle N_{test}\rangle^b$ | $\langle N_{correct}\rangle^c$ | $\langle A_{test}\rangle^d$ | $\langle\kappa\rangle^e$ |
| 0.97 (SD 0.03) | 1000 (SD 0) | 697 (SD 132) | 0.71 (SD 0.12) | 0.67 (SD 0.13) |

[a] $\langle A_{val}\rangle$; average validation accuracy.

[b] $\langle N_{test}\rangle$; average number of images in the testing sets.

[c] $\langle N_{correct}\rangle$; average number of images in the testing sets predicted correctly.

[d] $\langle A_{test}\rangle$; average accuracy in the test sets.

[e] $\langle\kappa\rangle$; average value of the Cohen's Kappa coefficient for predictions of the testing sets.

performance of the DNN models for each family, whereas micro-, macro- and weighted-average captures different aspects of the overall performance of the DNN models across the test datasets. Values listed under the *Support* column indicate the total number of predictions considers for a particular evaluation of any of the listed measures. We provide a detailed description of the expressions used to calculate these quantities in the Materials & Methods section: *Statistical analysis of Multi-class predictions*.

The macro-average calculation consider all classes as equally important. Using this measure, we obtained macro-averages for precision, recall, and F1-scores as 0.77, 0.71, and 0.71, respectively. On the other hand, a micro-average metric constitutes an average biased by class frequency, which is more pertinent to our application because the test datasets are not balanced. Using this metric, we obtained micro averages for precision, recall, and F1-scores of 0.71, 0.71, and 0.71, respectively. Both, macro- and micro-averages indicated a satisfactory performance of the constructed DNN models. Examination of the local measures show that the models were able to produce good predictions for all classes, as indicated by a local F1-scores above 0.5. However, even though values of the local F1-scores for classes 3, and 9 exceed 0.5, the DNN models had difficulties predicting both classes as indicated by local recall values below 0.5. Finally, the local measures indicated that we could satisfactorily predict the remaining classes.

In a similar experiment, we trained 40 DNN models using fingerprints colored based on a reduced amino acid alphabet. The trained models were used to predict the family lineage of 80,000 fingerprints with an average accuracy of 0.57. A statistical summary of these runs is presented in S4 Table. We use the statistical data in Tables 4 and S4 to compare the F1-score local for predictions for fingerprints produced with difference color schemes. Fig 5 shows a plot of the F1-score local metric as a function of the lineage family, for both types of DNN models. This plot shows that the DNN models trained with fingerprints colored using a residue-charge color scheme were generally able to predict the family lineages with the largest number of members better than the DNN trained on fingerprints derived from the reduced-residue type coloring scheme. Models using fingerprints obtained with the reduced-residue type coloring scheme performed poorly predicting families 1, 2, 9, and 10 as indicated by the local F1-scores below 0.5.

## Detection of binding site preferences of anti-EBOV GP antibodies

In many instances, the specificity of an Ab for certain epitopes determine the neutralization and protective properties of the antibody. For example, Abs targeting the fusion loop on the E protein of dengue virus have, in general, poor neutralizing capabilities even though their

**Table 4. Detection of a specific lineage family.** Summary statistics[a] of 80 DNN models used for classification of 28 antibodies belonging to ten family lineages using fingerprints colored according to the charge coloring code.

| Lineage family | Number of Abs[b] | Precision Local | Recall Local | F1-score Local | Support |
|---|---|---|---|---|---|
| 1 | 4 | 0.91 | 0.69 | 0.79 | 12000 |
| 2 | 4 | 0.61 | 0.99 | 0.75 | 12000 |
| 3 | 4 | 1.00 | 0.46 | 0.63 | 12000 |
| 4 | 3 | 0.84 | 0.92 | 0.88 | 12000 |
| 5 | 3 | 0.67 | 0.58 | 0.62 | 12000 |
| 6 | 2 | 0.98 | 0.76 | 0.86 | 12000 |
| 7 | 2 | 0.54 | 0.72 | 0.62 | 12000 |
| 8 | 2 | 0.77 | 0.66 | 0.71 | 12000 |
| 9 | 2 | 0.91 | 0.46 | 0.61 | 12000 |
| 10 | 2 | 0.51 | 0.87 | 0.65 | 12000 |
| micro average | | 0.71 | 0.71 | 0.71 | 12000 |
| macro average | | 0.77 | 0.71 | 0.71 | 12000 |
| weighted average | | 0.77 | 0.71 | 0.71 | 12000 |

[a] Values computed with the python Scikit-learn library for machine learning and statistical modeling [25].

[b] This corresponds to the number of Abs associated with the lineage family.

binding strength can be high [26]. Thus, it is critical to know the specific antigenic site recognized by an Ab. With this objective in mind, we decided to explore the use of DNN models for classification of Abs based on their epitope preferences. For this analysis, we selected a set of anti-EBOV antibodies that recognized epitopes in the EBOV glycoprotein (GP) trimer. Most of these epitopes have been well-characterized through structural studies and binding assays [23]. Fig 6 depicts the areas of the EBOV GP trimer targeted by these Abs. Based on their epitope preference, we identified three subsets, $Set_1$ (9 Abs), $Set_2$ (6 Abs), and $Set_3$ (15 Abs) as listed in S3 Table. Abs from $Set_1$ and $Set_2$ are KZ52 competitors [23] and recognize epitopes of Abs ADI-15734 and ADI-15878, respectively, that are located at the base of the EBOV GP trimer (Fig 6B). Abs from $Set_3$ are competitors of ADI-15974 and bind at the α-helical heptad

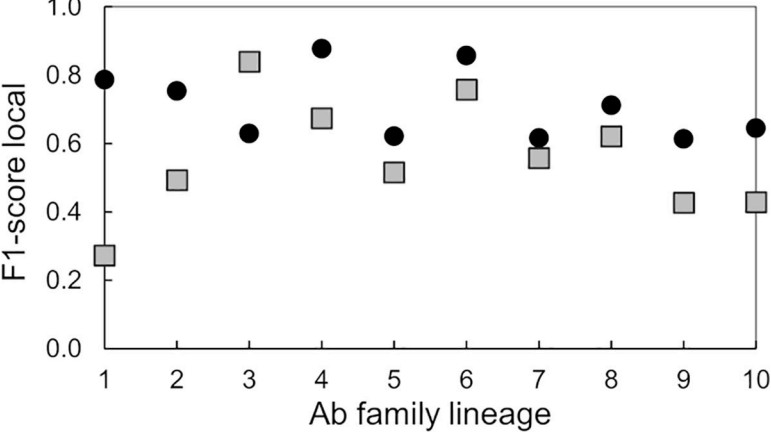

**Fig 5. Prediction accuracy of DNNs trained to detect Ab family lineage.** Plot of the "F1-score local"-metric as a function of the family lineage for two types of DNN models that we trained with Ab fingerprints generated by two alternative coloring schemes, i.e., by residue charge (black circles) or by reduced-amino-acid alphabet (grey squares).

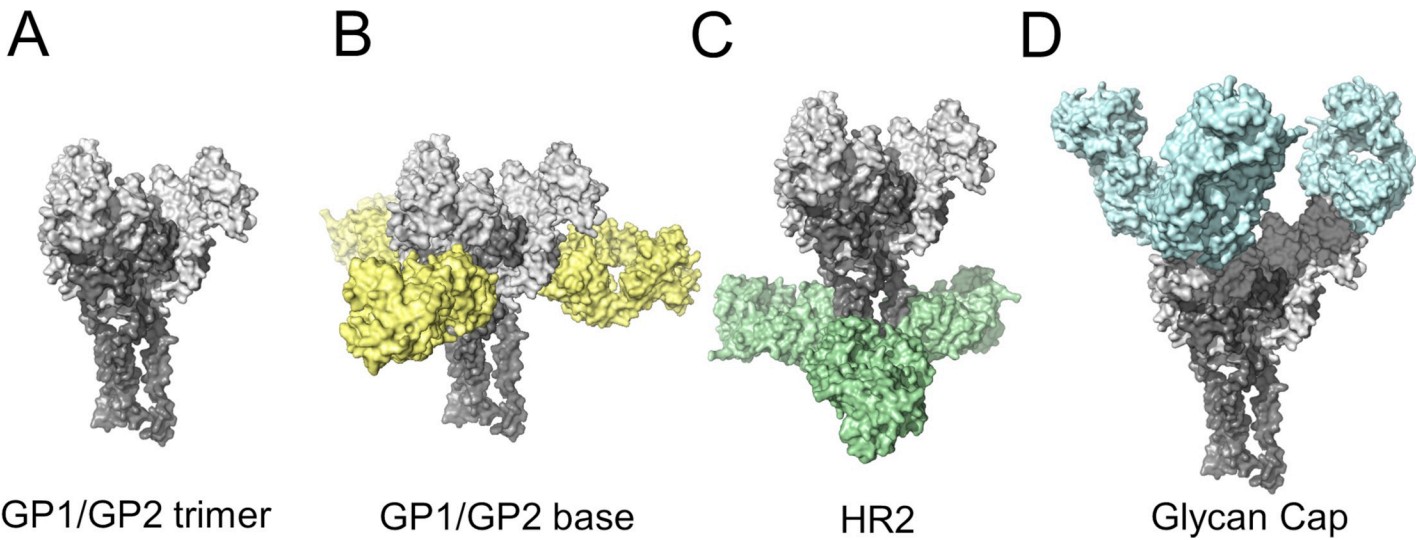

**Fig 6. Main regions of the EBOV GP trimer for Ab recognition.** (A) Structural model of the EBOV GP trimer recognized by anti-EBOV Abs. Abs have been colored according to the regions of the trimer that they bind, i.e., B) base of the trimer, C) the α-helical heptad repeat 2 (HR2) region, and D) the glycan Cap domains.

repeat 2 in the GP2 "stalk" (HR2) region (Fig 6C) region (See Materials & Methods Section: *Set of anti-EBOV antibodies*).

We carried out three sets of experiments using the anti-EBOV GP trimer Abs to differentiate Abs binding to two or three epitopes. For the first round of experiments ($Exp_1$), we chose Abs in $Set_1$ and $Set_3$ that belong to different competition groups [23]. For our second round ($Exp_2$), we selected Abs associated with $Set_1$ and $Set_2$. The binding epitopes of these Abs are near each other or may even overlap and are located at the base of the EBOV GP trimer. Finally, we carried out the third round of experiments ($Exp_3$) using the Abs from $Set_1$, $Set_2$, and $Set_3$ to assess specificity for one out of three possible binding sites.

To train the DNN models in $Exp_1$, we first selected fingerprints corresponding to 24 Abs ($Set_1$ and $Set_3$). We then built our training/validation sets by randomly selecting a subset of the 24 Abs, and used the remaining ones to construct a test set. Table 5 shows a summary of 30 DNN models trained on Abs fingerprint to detect the binding sites preferences on these Abs. We computed the average percentage of correct assignments as 79% (SD 13%), with a Cohen's

**Table 5. Classification of Abs recognizing two epitopes at either the base (ADI-15734) or the HR2/MPER region (ADI-15974) of the EBOV-GP trimer.** We colored the fingerprints for the DNN models according to the charge coloring code.

| No. DNN models | Training and Validation | | | Testing Results | | | |
|---|---|---|---|---|---|---|---|
| | $\langle N_{training}\rangle$ [a] | $\langle M_{validation}\rangle$ [b] | $\langle A_{val}\rangle$ [c] | $\langle N_{test}\rangle$ [d] | $\langle N_{correct}\rangle$ [e] | $\langle A_{test}\rangle$ [f] | $\langle \kappa\rangle$ [g] |
| 30 | 1007 (SD 62) | 282 (SD 28) | 0.98 (SD 0.02) | 501 (SD 85) | 395 (SD 91) | 0.79 (SD 0.13) | 0.67 (SD 0.22) |

[a] $\langle N_{training}\rangle$; average number of fingerprints selected for training.

[b] $\langle M_{validation}\rangle$; average number of fingerprints selected for validation.

[c] $\langle A_{val}\rangle$; average validation accuracy.

[d] $\langle N_{test}\rangle$; average number of images in the testing sets.

[e] $\langle N_{correct}\rangle$; average number of images in the testing sets predicted correctly.

[f] $\langle A_{test}\rangle$; average accuracy on the test sets.

[g] $\langle \kappa \rangle$; average value of the Cohen's Kappa coefficient for predictions of the testing sets.

**Table 6. Classification of Abs (ADI-15734 and ADI-15878) recognizing two epitopes at the base of the EBOV-GP trimer.** We colored the fingerprints for the DNN models according to the charge coloring code.

| No. DNN models | Training and Validation | | | Testing Results | | | |
|---|---|---|---|---|---|---|---|
| | $\langle N_{training}\rangle$ [a] | $\langle M_{validation}\rangle$ [b] | $\langle A_{val}\rangle$ [c] | $\langle N_{test}\rangle$ [d] | $\langle N_{correct}\rangle$ [e] | $\langle A_{test}\rangle$ [f] | $\langle\kappa\rangle$ [g] |
| 30 | 893 (SD 64) | 269 (SD 34) | 1.00 (SD 0.01) | 478 (SD 76) | 420 (SD 79) | 0.88 (SD 0.10) | 0.79 (SD 0.19) |

[a] $N_{training}$; number of fingerprints selected for training of the model.

[b] $M_{validation}$; number of fingerprints selected for validation.

[c] $A_{val}$; validation accuracy.

[d] $N_{test}$; number of images in the testing sets.

[e] $N_{correct}$; number of images in the testing sets predicted correctly.

[f] $A_{test}$; accuracy as evaluated using the test sets.

[g] $\kappa$; Cohen's Kappa coefficient for predictions of the testing sets.

Kappa coefficient of 0.67 (SD 0.22), indicating that the DNN model predictions are substantially different from random predictions.

The second experiment, $Exp_2$, was carried out using Abs from $Set_1$ and $Set_2$. Table 6 show the results from training, validating, and testing of 15 DNN models. The average number of correct predictions was 88% (SD 10%), and the computed Cohen's Kappa coefficient was 0.79 (SD 0.19).

Finally, we carried out a third experiment, $Exp_3$, to evaluate if DNNs can discriminate Abs based on their exclusive binding specificity for one out of three possible sites. We trained 20 DNN models using 30 Abs from $Set_1$, $Set_2$, and $Set_3$ listed in S3 Table. We used fingerprints colored according to the charge coloring code and no image enhancement during training. Table 7 provide a statistical summary of these calculations, showing an average of 71% (SD 9%) correct predictions, with a computed Cohen's Kappa coefficient of 0.55 (SD 0.17).

Table 8 provide additional statistical data quantifying the performance of the models in this three-class classification exercise. Local measures of precision, recall, and F1-scores, show that the DNN models are able to satisfactorily predict Abs from $Set_1$ and $Set_3$, but not from $Set_2$. Overall, the micro- and macro-averages of precision, recall, and F1-scores show an acceptable performance of the DNN models across the sets of test images.

## Detection of binding site preferences of HIV antibodies

After training DNN models for predicting the preference of a set of Abs for specific epitopes in EBOV GP trimer, we explored if this approach could be applied to other families of Abs. HIV is a rapidly mutating virus, and as a consequence, a difficult target for vaccine development. Research efforts on this field have focused on Abs capable of neutralizing multiple viral strains. A large number of broadly neutralizing Abs against HIV are well characterized, and extensive information on neutralizing antibody sequences and potencies, together with a substantial amount of structural data can be found in public databases such as CATNAP [27] and PDB [28]. Using these resources, we selected a set of 72 broadly neutralizing Abs that target specific epitopes on the HIV-1 GP120/GP41 protein complex. Using a series of 3-D experimental structures of Ab-GP120/GP41 complexes, we carried out multiple superpositions to identify the binding epitopes of thee Abs (additional information on this analysis is given in the Materials & Methods: *Set of anti-HIV antibodies*). We found that a majority of the broadly neutralizing Abs studied so far and shown in Fig 7 preferentially target two sites on GP120. One of the targeted epitopes mapped to the primary receptor CD4-binding region of HIV-1 GP120

 

**Table 7. Classification of 30 Abs that bind exclusively to one out three possible epitopes.** Two binding sites recognized by Abs from $Set_1$ (ADI-15734 competitors) and $Set_2$ (ADI-15878 competitors) are located at the base of the EBOV-GP trimer, and the third epitope recognized by Abs from $Set_3$ (ADI-15974 competitors) is located at HR2/MPER region. All DNN models are based on evaluating fingerprint images based on the charge coloring code.

| DNN Model | Training and Validation Image Set Sizes | | | Testing Results | | | |
|---|---|---|---|---|---|---|---|
| | $N_{training}$[a] | $M_{validation}$[b] | $A_{val}$[c] | $N_{test}$[d] | $N_{correct}$[e] | $A_{test}$[f] | $\kappa$[g] |
| 1 | 1466 | 434 | 1.00 | 730 | 486 | 0.67 | 0.43 |
| 2 | 1505 | 525 | 0.94 | 620 | 399 | 0.64 | 0.44 |
| 3 | 1352 | 408 | 0.99 | 900 | 812 | 0.90 | 0.88 |
| 4 | 1550 | 470 | 0.94 | 640 | 430 | 0.67 | 0.40 |
| 5 | 1402 | 418 | 0.87 | 840 | 552 | 0.66 | 0.43 |
| 6 | 1402 | 418 | 0.89 | 840 | 668 | 0.80 | 0.69 |
| 7 | 1548 | 472 | 0.98 | 650 | 415 | 0.64 | 0.43 |
| 8 | 1318 | 462 | 0.87 | 860 | 704 | 0.82 | 0.75 |
| 9 | 1381 | 409 | 0.83 | 810 | 519 | 0.64 | 0.43 |
| 10 | 1505 | 525 | 0.99 | 620 | 489 | 0.79 | 0.72 |
| 11 | 1568 | 392 | 0.99 | 680 | 577 | 0.85 | 0.81 |
| 12 | 1468 | 452 | 0.94 | 650 | 456 | 0.70 | 0.57 |
| 13 | 1528 | 452 | 0.99 | 690 | 442 | 0.64 | 0.44 |
| 14 | 1424 | 436 | 0.86 | 900 | 568 | 0.63 | 0.40 |
| 15 | 1694 | 376 | 0.99 | 840 | 723 | 0.86 | 0.82 |
| 16 | 1554 | 546 | 0.93 | 860 | 503 | 0.58 | 0.31 |
| 17 | 1624 | 406 | 1.00 | 880 | 630 | 0.72 | 0.60 |
| 18 | 1720 | 430 | 0.99 | 800 | 551 | 0.69 | 0.55 |
| 19 | 1664 | 416 | 0.90 | 860 | 557 | 0.65 | 0.47 |
| 20 | 1772 | 528 | 0.97 | 700 | 441 | 0.63 | 0.42 |
| Average (SD) | 1522 (126) | 449 (49) | 0.94 (0.05) | 769 (102) | 546 (113) | 0.71 (0.09) | 0.55 (0.17) |

[a] $N_{training}$; number of fingerprints selected for training of the model.

[b] $M_{validation}$; number of fingerprints selected for validation.

[c] $A_{val}$; validation accuracy.

[d] $N_{test}$; number of images in the test sets.

[e] $N_{correct}$; number of images in the test sets predicted correctly.

[f] $A_{test}$; accuracy as evaluated on the test sets.

[g] $\kappa$; Cohen's Kappa coefficient for predictions of the test sets.

**Table 8. Classification of 30 Abs that bind exclusively to one out three possible epitopes in the EBOV GP trimer.** Summary statistics[a] of 20 DNN models trained using fingerprints colored according to the charge coloring code.

| Ab Set | Precision Local | Recall Local | F1-score Local | Support |
|---|---|---|---|---|
| $Set_1$ | 0.95 | 0.52 | 0.67 | 5650 |
| $Set_2$ | 0.82 | 0.67 | 0.74 | 4040 |
| $Set_3$ | 0.59 | 0.93 | 0.72 | 5680 |
| micro average | 0.71 | 0.71 | 0.71 | 15370 |
| macro average | 0.79 | 0.71 | 0.71 | 15370 |
| weighted average | 0.78 | 0.71 | 0.71 | 15370 |

[a] Values computed with the Python Scikit-learn library for machine learning and statistical modeling [25].

 

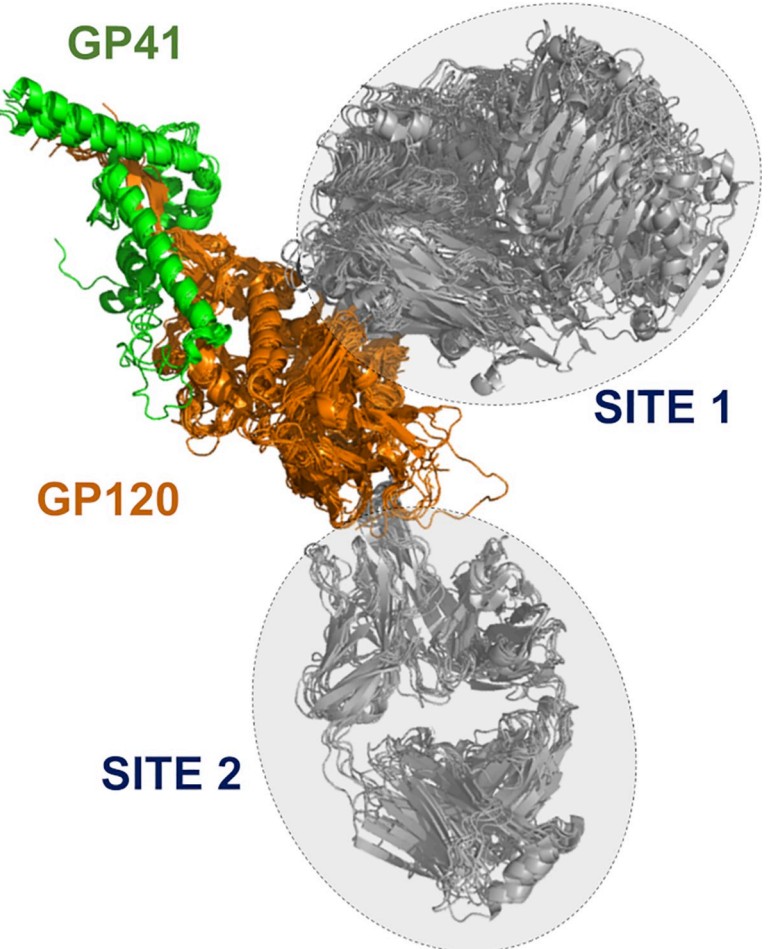

**Fig 7. Antibody recognition sites in HIV GP120/GP41.** Two main Ab binding regions based on the structural complex of the Ebola surface glycoprotein GP120/GP41 proteins with anti-HIV-1 Abs. Site 1 encompasses a structural overlay of 18 different site-specific Abs, whereas Site 2 contains 10 different Abs.

[29], whereas the second correspond to the V3 glycan region [30,31], associated with the envelope high-mannose patch that includes a glycan linked to ASN 332 in GP120.

We trained 50 DNN models using 12 to 18 Abs recognizing the CD4 binding site ($Site_1$) and 8 to 10 Abs binding to the V3 glycan site ($Site_2$). We carried out five experiments combining Abs from $Site_1$ and $Site_2$ using different ratios, as listed in the 2$^{nd}$ column of Table 9. For each experiment, we trained ten DNN models using different training, validation, and testing sets. To generate these sets we followed the protocol described in the Materials and Method Section: *Training and testing DNN for antibody classification on fingerprints* (see also Fig 3). Briefly, we used fingerprints derived from 56% to 75% of the Abs recognizing $Site_1$ for training and validation at an approximate 80/20 ratio. The fingerprints from the remaining Abs (25% to 44%) were kept separate and only used for testing. Similarly, 67% to 75% of the Abs binding to $Site_2$, were assigned to the training/validation sets, and their fingerprints distributed between training and validation sets at an approximate 80/20 ratio. The fingerprints of the remaining $Site_2$ Abs (25% to 33%) were only used for testing. Table 9 provide a statistical summary of the performance of 50 DNN models as evaluated on the test sets. We found that most of the trained DNN models were able to accurately predict new Ab classes based on fingerprint

**Table 9. Classification of Abs recognizing two distinct epitopes on the HIV-1 GP120/GP41 protein complex.** Statistical summary of five experiments in which we trained 50 DNN models using fingerprints colored according to the charge coloring code.

| Experiment | Ab assignment (Site 1/Site 2) | Training and Validation | | | Testing Results | | | |
|---|---|---|---|---|---|---|---|---|
| | | $\langle N_{training}\rangle^a$ (SD) | $\langle M_{validation}\rangle^b$ (SD) | $\langle A_{val}\rangle^c$ (SD) | $\langle N_{test}\rangle^d$ (SD) | $\langle N_{correct}\rangle^e$ (SD) | $\langle A_{test}\rangle^f$ (SD) | $\langle\kappa\rangle^g$ (SD) |
| 1 | 12 / 8 | 448 (23) | 171 (10) | 1.00 (0.0) | 362 (33) | 319 (52) | 0.87 (0.09) | 0.72 (0.28) |
| 2 | 14 / 8 | 519 (20) | 147 (5) | 1.00 (0.0) | 412 (25) | 345 (53) | 0.83 (0.11) | 0.57 (0.36) |
| 3 | 16 / 8 | 596 (23) | 215 (11) | 1.00 (0.0) | 364 (34) | 334 (43) | 0.92 (0.08) | 0.84 (0.15) |
| 4 | 16 / 9 | 554 (26) | 157 (7) | 1.00 (0.0) | 514 (34) | 439 (83) | 0.85 (0.14) | 0.73 (0.27) |
| 5 | 18 / 10 | 612 (26) | 173 (8) | 1.00 (0.01) | 588 (34) | 495 (53) | 0.84 (0.07) | 0.68 (0.13) |
| Averages of all models (SD) | | 546 (24) | 173 (8) | 1.00 (0.0) | 448 (32) | 386 (57) | 0.86 (0.10) | 0.71 (0.24) |

[a] $\langle N_{training}\rangle$; average number of fingerprints selected for training.

[b] $\langle M_{validation}\rangle$; average number of fingerprints selected for validation.

[c] $\langle A_{val}\rangle$; average validation accuracy.

[d] $\langle N_{test}\rangle$; average number of images in the testing sets.

[e] $\langle N_{correct}\rangle$; average number of images in the testing sets predicted correctly.

[f] $\langle A_{test}\rangle$; average accuracy as evaluated on the test sets.

[g] $\langle\kappa\rangle$; average value of the Cohen's Kappa coefficient for predictions of the testing sets.

similarities. The average accuracy, $\langle A_{test}\rangle$, over 50 models was 0.86 (SD 0.10), and the computed Cohen's Kappa coefficient was 0.71 (SD 0.24). These results indicate that these DNN models were effective in learning common features among the Abs of each class.

## Using one-class classification methods for Ab detection

The number of Abs in a repertoire having desirable properties such as high affinity for their cognate antigen or neutralizing activity against a given pathogen is usually very small. Although there are effective experimental methods to accomplish such tasks, they are time and labor intensive [23,32,33]. Hence, the development of computational methods for rapid screening of B-cell repertoires in search of those cells able to produce Abs with good characteristics is receiving considerable attention [16,34]. Due to the large diversity of the Abs in a repertoire, it is difficult to *a priori* classify them in a manageable number of classes for use in AI-based analyses. An efficient tool for Ab screening must be capable of learning features from a small set of Abs, and then search a much larger pool to identify other Abs with similar characteristics. This approach assume that the set of Abs with the desirable properties possess one or more features that distinguish them from the rest of the Ab repertoire. In this context, the computational problem is similar to those involving anomaly detections [35–38]. The goal in anomaly detection is to determine what instances in a dataset are truly different from all the others, i.e., identify the outliers. One-class classification (OCC) methods used in machine learning attempts to identify objects of a particular class from a much larger dataset of objects. In these methods, a classifier learns from a training set containing only samples of a single class. Deep learning methods for one class classification have been effectively used for anomaly detection. To explore the applicability of OCC for Ab detection, we have used the Robust Convolutional Autoencoder (RCAE) algorithm [35,39] for identification of Abs belonging to a single lineage from a much large set containing Abs from multiple families. A description of the RCAE and of the procedure to build the fingerprint datasets for training/validation and testing is provided in Materials and Methods Section: *One-class classification*.

Table 10 show a summary of twelve DNN models based on the RCAE method for OCC. The analysis shows high Area Under the Receiver Operating Characteristic (AUROC) values

**Table 10. Statistical summary of 12 DNN models trained to distinguish Abs from a specific family lineage using the Robust Convolutional Autoencoder one-class classification method.**

| DNN model | Training and Validation Image Set Sizes | | | | Testing Image Set & Results | | | |
|---|---|---|---|---|---|---|---|---|
| | $N_{training}$[a] | $M_{validation}$[b] | | AUROC Train[c] | $N_{test}$[d] | | AUROC Test[e] | normal Ab Id |
| | *normal* | *normal* | *anomalous* | | *normal* | *anomalous* | | |
| 1 | 765 | 135 | 5700 | 0.84 | 300 | 1500 | 0.95 | ADI-15841 |
| 2 | 765 | 135 | 5700 | 0.72 | 300 | 1500 | 0.65 | ADI-15916 |
| 3 | 765 | 135 | 5700 | 0.74 | 300 | 1500 | 1.00 | ADI-15925 |
| 4 | 765 | 135 | 5700 | 0.84 | 300 | 1500 | 0.79 | ADI-15785 |
| 5 | 765 | 135 | 5700 | 0.99 | 300 | 300 | 0.45 | ADI-15935 |
| 6 | 765 | 135 | 5700 | 0.89 | 300 | 300 | 0.81 | ADI-15772 |
| 7 | 765 | 135 | 5700 | 0.94 | 300 | 300 | 0.14 | ADI-15780 |
| 8 | 765 | 135 | 5700 | 0.61 | 300 | 300 | 1.00 | ADI-15784 |
| 9 | 442 | 78 | 3800 | 1.00 | 200 | 1000 | 0.99 | ADI-15843 |
| 10 | 510 | 90 | 3800 | 1.00 | 120 | 1000 | 0.89 | ADI-15912 |
| 11 | 442 | 78 | 3800 | 1.00 | 200 | 1000 | 1.00 | ADI-15978 |
| 12 | 442 | 78 | 3800 | 0.84 | 200 | 1000 | 0.98 | ADI-15861 |
| Average (SD) | | | | 0.87 (0.13) | | | 0.80 (0.27) | |

[a] $N_{training}$; number of fingerprints from normal Abs selected for training of the model.

[b] $M_{validation}$; number of fingerprints from normal and anomalous Ab classes selected for validation.

[c] AUROC Train is computed on the training set using the Python Scikit-learn library for machine learning and statistical modeling [25].

[d] $N_{test}$; number of fingerprints from normal and anomalous Ab classes in the testing sets.

[e] AUROC Test is computed on the testing set using the Python Scikit-learn library for machine learning and statistical modeling [25].

on the training sets with an average over all models of 0.87 (SD 0.13). Average AUROC metrics for application to the independent testing sets was reduced to 0.80 (SD 0.27). Despite the fact that training of these models was generally successful, the performance of three models (2, 5, and 7) was poor when applied to fingerprints from Abs not seen during training. Panel A and B of Fig 8 shows 20 images classified in the normal and anomalous classes produced by the 3[rd] DNN model in Table 10. The figure shows that the model was able to clearly separate the fingerprints from ADI-15925 that belongs to lineage 1 (i.e., the normal class) from fingerprints from all other families of antibodies defined as outliers. Similar results were obtained for DNN models listed as 1, 8, and 12 in Table 10.

To further assess the discriminative power of the DNN models for one-class classification, we produced a second batch of models using a larger set of Abs. Table 11 shows statistics for thirteen additional models trained to distinguish fingerprints of Abs from a single lineage from a large pool of fingerprints. The latter sets of fingerprints were derived from 304 Abs from the set of Bornholdt *et al.* [23]. Training sets for DNN models 2 to 13 contained more than 10,000 fingerprints from 242 or 243 anomalous Abs selected randomly from the larger set of 304. We excluded the remaining fingerprints from 61 Abs from the training sets and used them only as the anomalous class in the independent testing evaluation of the models. The number of Abs used for training and testing sets of the DNN models 1 and 2 was smaller than that used for the other models. We trained the models with a varying number of fingerprints for the normal Abs ranging from 500 to 900. The analysis presented in Table 11 shows an average AUROC value for the training sets of 0.94 (SD 0.14). Similarly, the average AUROC for applications of the DNN models to the independent testing sets was 0.95 (SD 0.07). With exception of model 13, the models were highly successful in discriminating fingerprints from Abs of the normal lineage from those belonging to anomalous Abs.

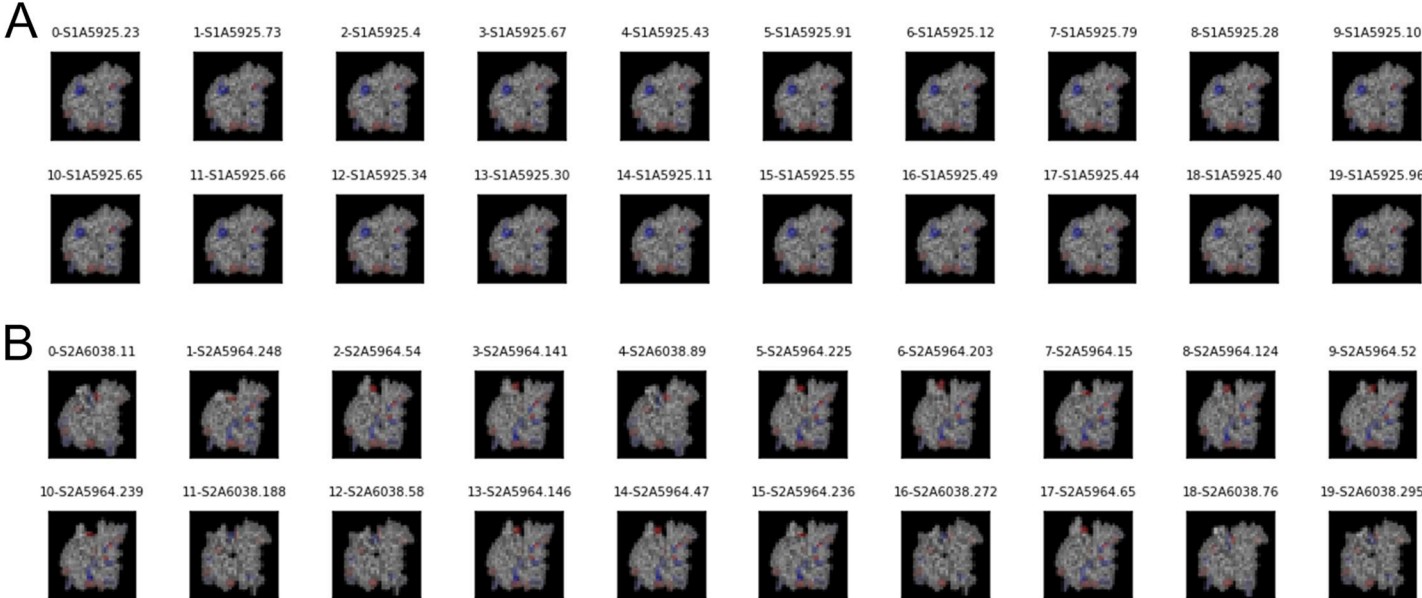

**Fig 8. Representative Ab fingerprints from normal and outlier classes.** (A) Exemplar images of Ab fingerprints classified as belonging to lineage 1 by one DNN model. The images from panel A are ordered according to scores assigned by the neural network. They are organized in rows of 10 images, with scores decreasing from left to right. (B) Exemplar images of Ab fingerprints classified as belonging to outliers (Abs from lineages other than 1) by the DNN model. The images from panel B are ordered according to scores assigned by the neural network. They are organized in rows of 10 images, with scores increasing from left to right. These results correspond to the DNN model listed as "3" in Table 10.

In a third application of the OCC method, we assessed if DNNs trained with fingerprints from a small subset of clonally-related Abs that recognize a particular epitope could detect Abs from different clonotypes with similar functionality. To carry out this study, we used the set of EBOV Abs from Bornholdt *et al.* [23], together with binding data presented in S5 Table from studies that associate the bulk of the Abs to specific competition groups. We selected twelve (12) Abs from the same lineage that were identified as competitors of KZ52 as the normal class (see S6 Table). In addition, approximately 180 Abs from the Bornholdt *et al.* [23] list that do not compete with Ab KZ52 were associated with the anomalous class and divided for validation and testing purposes at an approximate ratio of ~70/30 percent. Another 41 Abs listed as KZ52 competitors, but clonally distinct from the 12 "normal" Abs selected for training, were set aside for use in the testing sets (regular tests). We generated the test sets combining fingerprints of one "normal" Ab with images from ~50 Abs from the "anomalous" category using 100 fingerprints for each Ab. Thus, each testing sets contained 100 fingerprints from one "normal" Ab combined with ~5K fingerprints from various "anomalous" Abs.

In addition, we generated 59 "negative" test sets. To produce a negative test, we temporarily extracted an Ab from the "anomalous" class, changed its tag "normal," and added its fingerprints to the test set as representative of the "normal" class. As we did in the regular cases, we trained a DNN using the 12 Abs from the same lineage as the "normal" class, and a large number of KZ52 non-competitor Abs as the "anomalous" class. Over all, we trained 100 DNN models an evaluated their ability to detect from the test sets an Ab with binding properties similar to those of Abs from the normal training class. The Ab tagged as "normal" included in the test sets that was either a KZ52 competitor (regular test), or a KZ52 non-competitor (negative test). The evaluation of these test sets by the DNN models can lead to few different outcomes. We note that the set of Abs competitors of KZ52 contain various Abs binding to epitopes that

**Table 11. Statistical summary of 13 DNN models trained to distinguish Abs from a specific family lineage from a large Ab set using the Robust Convolutional Auto-encoder (RCAE) one-class classification method.**

| DNN model | Training and Validation Image Set Sizes | | | | Testing Image Set & Results | | | |
|---|---|---|---|---|---|---|---|---|
| | $N_{training}$ [a] | $M_{validation}$ [b] | | AUROC Train[c] | $N_{test}$ [d] | | AUROC Test[e] | normal Ab Id |
| | *normal* | *normal* | *anomalous* | | *normal* | *anomalous* | | |
| 1 | 765 | 135 | 7580 | 1.00 | 300 | 2000 | 0.98 | ADI-15925 |
| 2 | 765 | 135 | 12220 | 0.98 | 300 | 1510 | 0.97 | ADI-15784 |
| 3 | 765 | 135 | 14210 | 0.94 | 300 | 1940 | 1.00 | ADI-15861 |
| 4 | 765 | 135 | 15300 | 0.98 | 300 | 1850 | 0.92 | ADI-15841 |
| 5 | 442 | 78 | 10760 | 1.00 | 150 | 2340 | 0.99 | ADI-15978 |
| 6 | 765 | 135 | 14500 | 0.99 | 300 | 1850 | 0.96 | ADI-15925 |
| 7 | 765 | 135 | 13960 | 1.00 | 300 | 1990 | 0.99 | ADI-15916 |
| 8 | 765 | 135 | 14250 | 0.94 | 300 | 1900 | 1.00 | ADI-15785 |
| 9 | 765 | 135 | 12770 | 1.00 | 300 | 2280 | 1.00 | ADI-15935 |
| 10 | 765 | 135 | 14380 | 1.00 | 300 | 1870 | 0.91 | ADI-15772 |
| 11 | 765 | 135 | 15220 | 1.00 | 300 | 1730 | 0.99 | ADI-15780 |
| 12 | 442 | 78 | 10510 | 0.94 | 150 | 2390 | 0.87 | ADI-15843 |
| 13 | 510 | 90 | 10530 | 0.49 | 120 | 2370 | 0.77 | ADI-15908 |
| Average (SD) | | | | 0.94 (0.14) | | | 0.95 (0.07) | |

[a] $N_{training}$; number of fingerprints from normal Abs selected for training of the model.

[b] $M_{validation}$; number of fingerprints from normal and anomalous Ab classes selected for validation.

[c] AUROC Train is computed on the training set using the Python Scikit-learn library for machine learning and statistical modeling [25].

[d] $N_{test}$; number of fingerprints from normal and anomalous Ab classes in the testing sets.

[e] AUROC Test is computed on the testing set using the Python Scikit-learn library for machine learning and statistical modeling [25].

are different from the one recognized by the Abs in the family lineage used as the normal class. For such a reason, the expectation was that trained DNN models would detect correctly only a few Abs as the normal from the set of KZ52 competitors. Any Abs from the list of KZ52 competitors detected correctly as normal, would likely bind the same EBOV GP epitope recognized by the normal Abs from the lineage family. On the other hand. DNNs models applied to the negative test set should detect no normal Ab, since those tagged as normal belong to the set of KZ52 non-competitor Abs.

Given that our test sets were all skewed toward the anomalous class of Abs, we resorted to two different metrics to assess the performance of the DNN models. First, we used the AUROC value for the test set reported by that the RCAE method as a scoring function to produce a preliminary ranking of the DNNs performance. S7 Table lists the ten DNN models with the highest AUROC scores, together with additional statistical data associated with training of these models. The six highest scores in S7 Table were obtained for test sets containing a single KZ52 competitor as the normal class, producing AUROC values close to one. The remaining three DNN models (i.e., models 7, 9, and 10 in S7 Table) produced high AUROC values for their test sets (negative test), even though they only contained "anomalous" Abs.

As a second metrics, we used the decoding errors of fingerprints included in the test set. The DNN ranks the fingerprints according to the decoding errors (see Eq 3) with the best representatives of the normal class having the smallest error values. We should note that a DNN model always produces a classification of the fingerprints, irrespectively of the present of normal Abs in the test set. Thus, if the DNN produces the lowest decoding errors for fingerprints of an Ab labelled "anomalous," we reach the conclusion that the DNN was unable to detect the normal class among the pool of fingerprints. On the other hand, a DNN model has a successful

detection when it ranks at the top of the scale multiple fingerprints of the Ab in the test set carrying the "normal" tag. Because, by construction, our test sets have only 100 fingerprints of each Abs, we say that the DNN detected the normal Ab when many of its fingerprints are ranked among the top 100.

For cases where we evaluated the performance of a DNN using negative tests, however, if the errors of the fingerprints of the Ab (erroneously) tagged "normal" are found among the hundred smallest errors, the detection is considered a False Positive (i.e., an anomalous Ab detected as normal). Lastly, if the Ab with the normal tag included in the test set is a KZ52 competitor, and a DNN does not detect it, this does not necessarily represent a failure. The list of KZ52 competitor has Abs that bind to other epitopes beyond that recognized by Abs in the family lineage used for training.

Fig 9 groups separates the DNN models listed in S7 Table, based on the type of testing sets used, regular or negative, and provides an evaluation of the ten DNN models based on the decoding errors of the fingerprints. In column 4 of Fig 9, we display a series of graphs of the fingerprint decoding errors ordered from low to high values. Colored circles indicate errors associated with fingerprints from the Ab tagged "normal" in the test sets. Multiple colored circles among the lowest 100 errors is an indication that the DNN detected the normal class. Column 5 of Fig 9 describes the final evaluation of the DNN models.

We found that DNN models 1, 2, 4, and 6 in Fig 9, ranked 90, 75, 40, and 35 fingerprints of KZ52 competitors A15877, A1598, A15741, and A15952, respectively, among the 100 "most normal" images, with some of the images having the lowest decoding errors. We considered that these DNN models successfully detected KZ52 competitors as the normal class. DNN models 3 and 5, assigned the lowest decoding errors to fingerprints from "anomalous" Abs. These DNN models, however, ranked among the 100 "most normal," 35 and 6 fingerprints, respectively, from KZ52 competitors A16005 and A15935. Thus, these results could be considered as partial detections. The remaining DNN model (#8) used in a regular test did not list any of the fingerprints of the Ab tagged as "normal, thus, it was considered a non-detection.

Lastly, the three DNN models with high AUROC scores that were used to evaluate negative test sets (i.e., DNN models 7, 9, and 10), did not listed errors of fingerprints from the decoy "normal" Ab among the 100 lowest. Thus, we evaluated their performance non-detections.

Based on this analysis, we conclude that DNN models 1, 2, 4, and 6 have correctly detected Abs from the group of KZ52 competitors as likely candidates of Abs sharing the binding properties of the Abs from the same lineage used for training purposes. Two other DNN models, 3 and 5, only produced partial detections of Abs A16005 and A15935, which are considered less likely candidates.

By construction, the six Abs detected do not belong to the same lineage of the Abs form the normal class used in training, Lineage annotation and CDR sequencing information for these Abs is given in S6 Table under the subheading *"Antibodies detected."*

Independently, we collected a series of experimental structures of anti-EBOV Abs in complex with GP1,2 from PDB [28] and the PDBe Electron Microscopy Database [40]. Based on 3-D-comparisons of these Ab complexes, we considered 11 different epitopes in EBOV GP1,2 trimer, ten of these epitopes are located at the base of the trimer and one is found in the glycan cap. We carried out a sequence comparison of the Abs detected by the DNNs in this experiment, with the eleven experimentally determined Abs [23,41–48]. We found that Ab ADI-15734, which belongs to the set of normal Abs used for training purposes, shows the best percentages of sequence identity with the detected Abs, with the exception of ADI-16005, as shown in S8 Table. These results may be considered as an additional indication that the detected Abs shared similar binding preferences with Abs from the lineage family used to train the DNNs

| Results from Regular tests | | | | |
| --- | --- | --- | --- | --- |
| Rank[a] | Ab ID | Number of top 100 images | Fingerprint errors[b] | Evaluation |
| 1 | 15877 | 90 | | Detection |
| 2 | 15958 | 75 | | Detection |
| 3 | 16005 | 35 | | Partial detection |
| 4 | 15741 | 40 | | Detection |
| 5 | 15935 | 6 | | Partial detection |
| 6 | 15952 | 35 | | Detection |
| 8 | 15865 | 0 | | No detection |
| **Results of Negative tests [c]** | | | | |
| 7 | 15886 | 0 | | No detection |
| 9 | 15925 | 0 | | No detection |
| 10 | 15820 | 0 | | No detection |

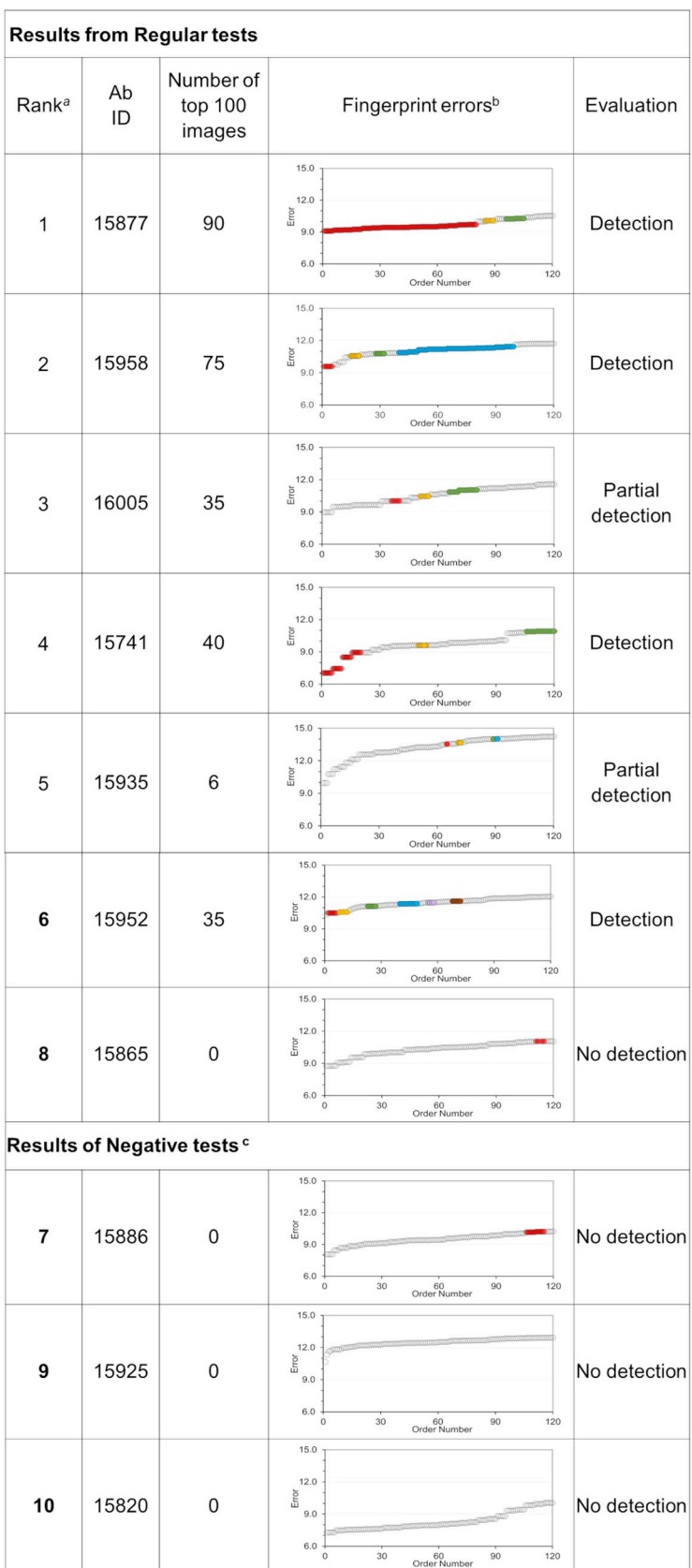

**Fig 9. Detection of clonally diverse antibodies using the OCC method RCAE.** [a] This number corresponds to the ranking assigned to the 100 DNN models based on the AUROC score computed on the testing set. [b] Image reconstruction errors ranked from low to high. Gray circles are associates with fingerprints from anomalous Abs. Colored circles highlight clusters of errors for fingerprints of the Abs from the "normal" class. Note that the graphs only display the reconstruction errors of 120 fingerprints from each testing set. [c] The test sets used to evaluate the DNN models below contained only Abs that do not compete with KZ52 in an attempt to detect false positives (i.e., the Ab representing the normal class was a decoy).

## Explanation of DNN classifications using an interpretable model

To verify that the DNN predictions relied on shape and amino-acid properties of the generated fingerprints, we used the program LIME (see the Materials & Methods Section: *Explaining DNN predictions with LIME)* to analyze the predictions from our DNN models. LIME is an algorithm that provides a realistic explanation for the prediction of a classifier by approximating the prediction locally with an interpretable model. For this particular exercise, LIME was used to detect the most relevant features of a fingerprint that a trained DNN model identifies to produce the association (i.e., the prediction) of the fingerprint with one of the Abs classes under consideration. We trained a DNN model to distinguish fingerprints from EBOV Abs that bind exclusively to one out three possible epitopes, as described earlier in Section "*Detection of binding site preferences of anti-EBOV GP antibodies.*" The resulting DNN model predicted the type of Ab associated with fingerprint from a particular testing set with 80% accuracy. To carry out the LIME analysis, we selected from the testing set a group of fingerprint images for each of three Ab types considered. The fingerprints selected corresponded to positive predictions of the DNN model. The results of this analysis is presented in Fig 10 as a series of related images highlighting the contributions of pixels in the fingerprints that the DNN considers during the decision making process. The images under the *fingerprint* column correspond to the actual fingerprint evaluated by the DNN model. Columns 1 and 5 identify the group of the Ab associated with the fingerprint, as described in S3 Table). Columns labelled *Explanation Top Class Pros-Cons* show composite images of the set of pixels having the largest positive (green) and negative (red) contributions to the predictions, superposed onto of the original fingerprints. The columns labelled *Heatmap* show heat maps where every pixel is colored according to its contribution to the prediction, with pixels with the largest positive contributions in dark blue and pixels with the most negative contributions in dark red.

Analysis of these images show that the most relevant features that the DNN model selects to produce a correct prediction map consistently to similar areas on the fingerprints of the same type of Ab. Furthermore, the areas containing selected features, while they may partially overlap, they are markedly different for fingerprints from Abs belonging to different types. Finally, we note that the main features contributing to the predictions are not constraint to areas on the fingerprints associated with CDR regions from the heavy chain, but important contributions originate from region mapping to the Ab light chain.

We carried out a similar exercise using the DNN model trained to discriminate HIV Abs that bind to two different sites (see Section "*Detection of binding site preferences of HIV antibodies*" above). This DNN model was able to correctly predict the binding site preference of the associated Abs. Using a testing set of 329 fingerprints from Ab not used for training purposes, the DNN model achieved an accuracy of 97%. Fig 11 shows LIME results for four arbitrarily chosen fingerprints belonging to the testing set. These results indicate that the DNNs decisions are based on features that capture charges in the Ab binding site regions together with regions with no net charge. Pixels with important contributions are also associated with regions that define the borders of the Ab binding site region associated with structural features derived from the 3-D models.

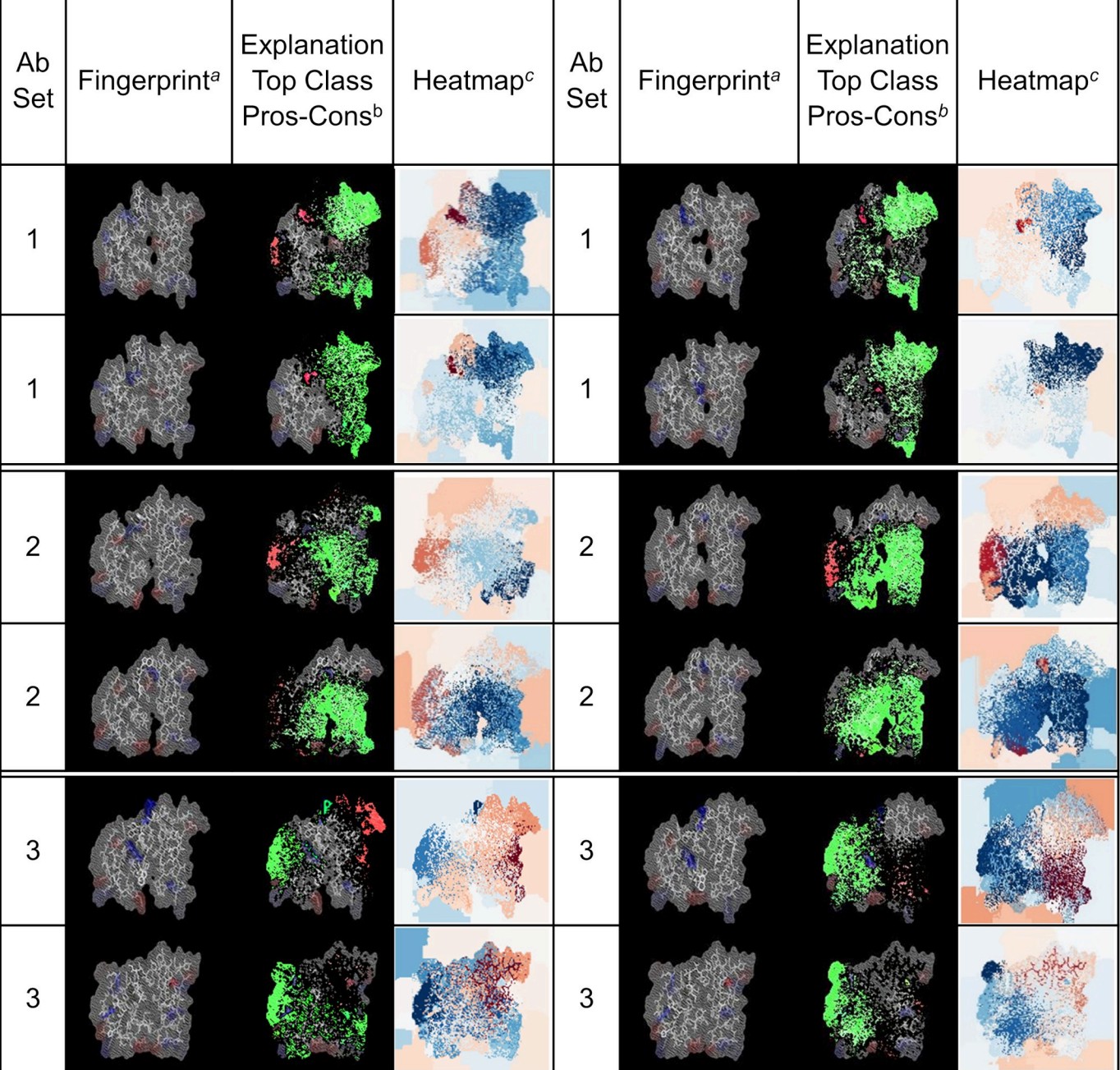

**Fig 10. LIME analysis evaluating the reliability of predictions from a trained DNN model.** [a] The column lists images of arbitrary fingerprints associated with the Abs listed under the "Abs Set" column. [b] The column contains images generated as the superposition of three elements, a) green color represent the most relevant pixels used by the DNN to generate the prediction, i.e., those shown in the image from column 3; b) bright red pixels have the most negative contribution to the prediction; and c) the remaining pixels from the original fingerprint image. [c] This column contains heatmap images describing the contribution of each pixels on the fingerprint to the prediction generated by the DNN model. The color scale ranges from dark blue for the most relevant contributions to dark red for the most negative ones. The color scale is selected independently for each heatmap based on the scores assigned by LIME.

## Discussion

In this work, we used a combination of artificial intelligence methods for image recognition together with computational tools for Ab-structure modeling to develop the technical foundation for analyzing B-cell receptor high-throughput sequencing data. The ultimate goal is to

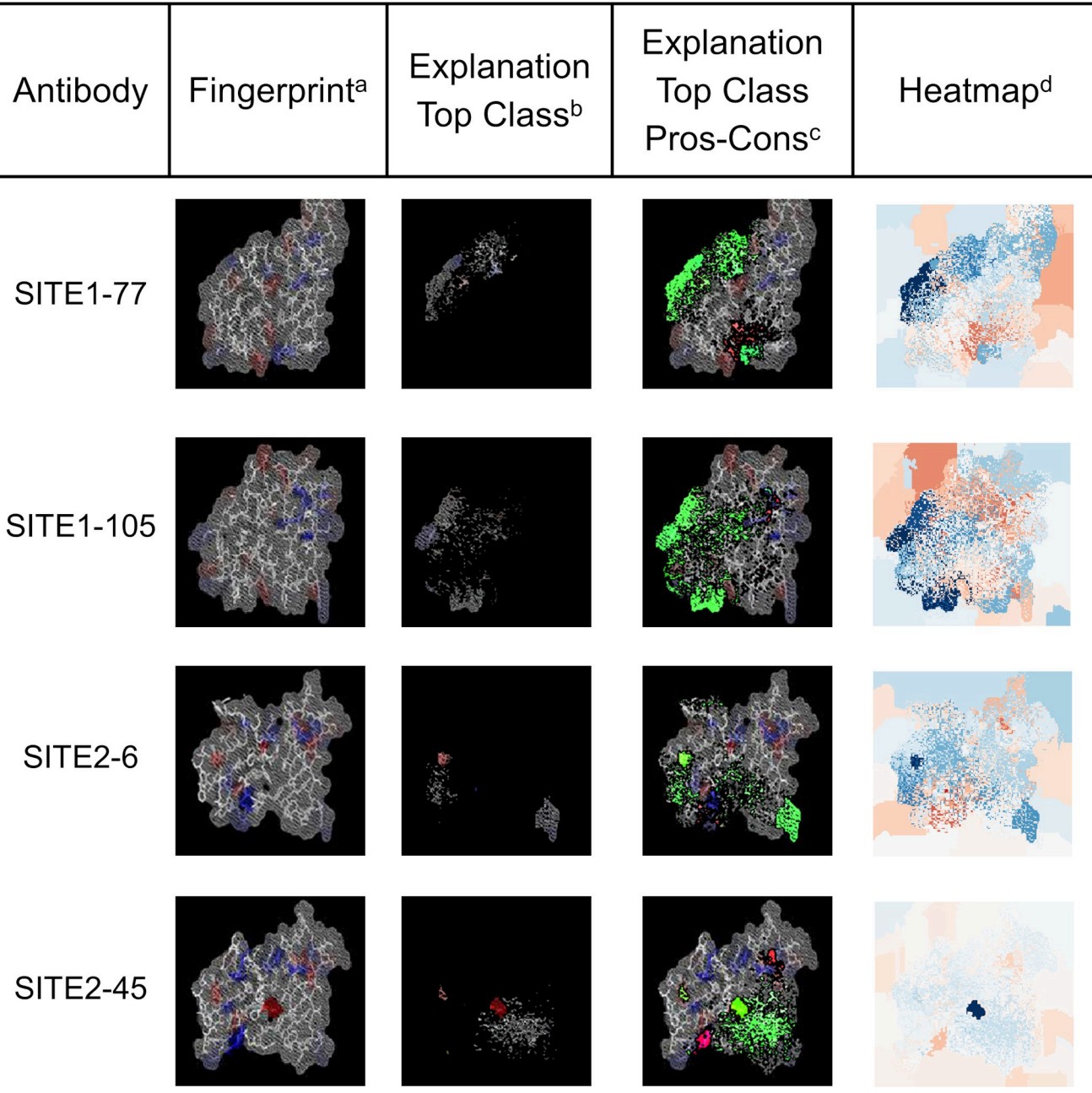

**Fig 11. LIME analysis evaluating the reliability of predictions of a DNN model trained for classification of HIV Abs based on their binding preference.** [a] The column lists images of arbitrary fingerprints associated with HIV Abs binding to SITE1 and SITE2 as defined in Fig 7. [b] Images in this column show to the most relevant pixels from the analyzed fingerprint used by the DNN model to generate the associations with the correct Ab. [c] The column contains images generated as the superposition of three elements, a) green color represent the most relevant pixels used by the DNN to generate the prediction, i.e., those shown in the image from column 3; b) bright red pixels have the most negative contribution to the prediction; and c) the remaining pixels from the original fingerprint image. [d] This column contains heatmap images describing the contribution of each pixels on the fingerprint to the prediction generated by the DNN model. The color scale ranges from dark blue for the most relevant contributions to dark red for the most negative ones. The color scale is selected independently for each heatmap based on the scores assigned by LIME.

develop a computational method for enriching pools of known disease-specific antibodies through the detection of clonally distinct Abs with similar functionality. In particular, cases where such common functionality cannot be inferred from sequence based analysis. With this objective in mind, we are exploring the use of DNNs to detect common key features on the

Abs. We designed a series of applications where we trained DNNs to predict functional characteristics of the Ab, using two-dimensional images of the Ab binding site. No explicit sequence or residue-connectivity information is used during training of the DNN models. These images or fingerprints correspond to reduced representations of Abs that highlight particular properties of the residues forming the binding site such as the net charge distribution using a simple color scheme. It must be noted that the current method is not intended to address problems such as the prediction of single mutation that disrupt binding or destabilize Ab-antigen complexes.

We first showed that training DNN models to carry out the task of classifying images of two different Abs was robust and highly successful with an average success rate of 96% (SD 5.5%) when used in test sets containing fingerprint images of Abs that were not used for either training or validation.

Next, we trained and assessed DNN models to identify Abs from the same B-cell lineage or family. Abs belonging to the same family lineages retain common residue patterns in their Ab binding site region, patterns that a DNN can learn and use for classification. However, metrics associated with each particular lineage showed that Abs from only three lineage families (i.e., 1, 2, and 4) where predicted with high confidence from the fingerprints, while the DNN models encountered difficulties predicting Abs associated with one of the lineages family (i.e., 9). Although the performance was acceptable based on the overall statistics for precision, recall, and F1-scores (Table 3), the analysis highlighted the presence of lineage dependence in the prediction models. These results imply that the resolution of the model prediction is dependent on certain molecular changes that define the details of the Ab binding site region, such as the replacement of a charged residue by a neutral one, or one with opposite charge, as it occurs in family 3 (see changes in the centers of the fingerprints for this family, shown in S2 Fig). In other cases, the molecular changes seem to be more subtle–either the changes at the Ab binding site region are small enough not to register in the 2-D projection or there might be additional conformational considerations not capture by Rosetta-generated structural ensemble that may be due to differences in the template structures selected for 3-D model generation. The limited number of Abs in each lineage may also influence the results. The limited number of Abs in each lineage may also influence the results. For example, family lineages 6 to 10 consist of two Abs each, and the number of images used for training and validation may have been under-represented with respect to the remaining classes.

When adapted to discriminate between two possible epitope specificities, our DNN models classified Abs against EBOV-GP and HIV-1-GP140/GP120 epitopes with average accuracies ranging from 71% to 88%. The DNN models classified Abs recognizing one out of three epitopes in EBOV-GP with an average accuracy of 71%. As we pointed out earlier, none of the Abs present in the test sets where use for training the DNN models. Collectively, fingerprint-based DNN image-recognition models trained on anti-EBOV and anti-HIV-1 Ab binding sites indicated that they could perform as classifiers for detection of epitope specificity.

We also verified that the shape and amino-acid property features from the fingerprints underlie the predictions of the trained DNN models. Application of the LIME methodology showed that the decision-making process used by the DNN models were based on groups of pixels associated with one or more regions of the Ab binding sites. These regions are associated with charged residues, but also include portions of the Ab binding site surface where net charges are absent, i.e., polar and/or non-polar surfaces. Other pixels associated with the high weight or explanatory power are linked to regions delimiting the Ab binding sites (boundary shape), and indicate that the overall structural shape of the Ab binding surface is also an important factors contributing to the DNN decision.

Finally, our explorations on identifying specific Abs from a larger pool of Abs were encouraging. We found that application of the RCAE method for one-class classification could be implemented for training DNN models using fingerprints from Abs belonging to a specific lineage. The final DNN models were able to identify fingerprints of an Ab from the same lineage from a larger group of fingerprints made up from unrelated Abs, despite that none of the fingerprints in the testing set corresponded to Abs used during training and validation of the models. We also showed that the RCAE method was successfully applied to train DNNs on a small set of Abs from the same clonotype with similar binding properties. The trained DNN models were able to detect a handful of Abs with similar binding properties but larger sequence diversity than that of the set used for training.

## Methodological considerations and limitations

Apart from the readily available amino acid sequences of Abs, application of the current methodology requires additional information on the functional properties of the Abs that is to be captured and classified in the DNN models. This may include Ab-antigen complex specific information such as alanine-scanning mutagenesis and binding competition assays for a number of "related" Abs. This information is required for an effective selection of Abs used during the training stage of the DNN models but may not always be readily available due to lack of extensive experimental data. We must note that this methodology does not use Ab lineage information and sequence information is only used to produce the structural models of the Abs to determine likely distributions of key residues. The Ab sequences are not used to train the DNN models.

It is worth noting that Abs are multifunctional biomolecular complexes and their different properties can be associated with different region of their 3-D structure. Properties such as epitope specificity and binding affinity are directly related to the structural features and the amino acid composition of the Ab binding site region. Hence, these types of properties can be investigate with the methodology presented here. On the other hand, there are other properties or functions of Abs that do not depend solely on the structure of the Ab binding site region, such as the neutralization capacity of Abs, which has been shown to depend not only on the binding strength of the Ab, but also on the number of accessible epitopes on the antigen surface [49]. Analysis of Ab functions that do not involve the Ab binding site region such as cell signaling that require the Fc region of Abs are also out of the scope of the current methodology.

We found that the partition of Abs and their fingerprints into training, validation, and testing sets was an important issue to incorporate into the method development. The performance of the DNN models was greatly affected by the available number of Abs of a given class, particularly when the number of Abs of a given category is small. For the classification of Abs by their epitope specificity, we found that the Abs added to the training set needs to be carefully considered, such that the training set contains Abs that recognize a representative set of similar or overlapping epitopes.

Intuitively, we expected that DNN training using fingerprints based on a reduced-residue code would improve the predictive capacity of the models given that such fingerprints contain additional information regarding other types of residues beyond the charge ones. While we do not have a clear explanation for the lack of improvement of the predictions, the new fingerprint color patterns due to the additional residue types contribute noise in terms of information overflow that weakens the actual signal instead of enhancing it.

The developed methodology was able to capture common patterns among two-dimensional representations of the Ab binding site regions displaying the position of key residues. The DNN models were able to learn these patterns and detect similarities in fingerprints from

unseen Abs. The reduced representation of the Ab binding site regions seems to incorporate a spatial component that help the models discriminate among those patterns. DNN models trained for recognition of the epitope preference of Abs proved effective, similarly, the application of the RCAE method for identification of Ab from a single category performed well.

We observed that some of the solutions of the optimization problem did not lead to DNN models with good predictive capacity. One of the causes of these failure relate to the weights of a trained DNN model determined through an optimization process of a multidimensional loss or cost function. Inherent to these optimization schemes is a stochastic procedure that typically converges to one of many possible local minima, hence, the true minimum cannot be guaranteed.

Another contribution to the failure of some DNN models related to balance of the training sets. The number of Abs associated with different classes can be quite diverse, leading to a large imbalance in the number of fingerprints use for training and validation. Hence, the final DNN models produced after the training process tend to be biased toward the most populated class.

We also attempted to train DNNs for classification of Abs against the EBOV GP trimer using a broad definition of *epitope* or *binding site*. We divided the set of Abs from Bornholdt *et al.* [23] into three sets that recognize the GP1/GP2 base of the trimer, the glycan cap, and HR2 regions, respectively. However, our attempts to train DNN models to classify Abs based on these broader categorical definitions were not consistently successful.

The validation accuracy ($A_{val}$) reported in this work is a metric associated with the quality of the DNN model that measures its accuracy on the validation set. Evaluation of $A_{val}$ occurs at the end of the optimization cycle defined by an epoch. We explored if this measure retains any relation with the actual performance of the DNN model in classifying fingerprints from the independent test sets. In principle, there is no guarantee that a DNN model that performs well in the validation set, i.e., with a high $A_{val}$, will generate successful predictions for Abs not previously seen by the model. In Fig 12, we plot the percentage of correct predictions in independent test sets and the associated Cohen's Kappa coefficient as a function of the validation accuracy of the DNN models used for detection of an Ab family lineage listed in Table 2. The horizontal dashed line at κ equal 0.4 was drawn based on Cohen's interpretation of Kappa

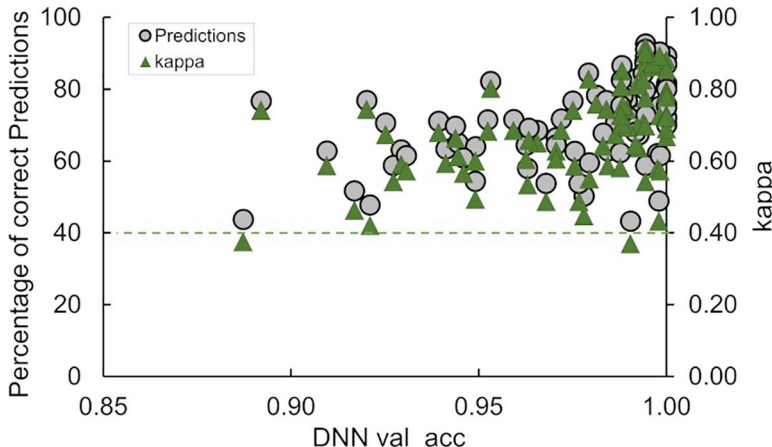

**Fig 12. Training of DNN for recognition of Abs from ten lineages.** The validation accuracy ($A_{val}$) is a metric associated with the quality of the DNN model that measures the accuracy on the validation set and is potentially prone to overfitting. The green horizontal line at κ equal 0.4 divides the set of predictions on independent test sets of fingerprints into significant ($\geq 0.4$) and no-significant ($< 0.4$).

[50,51]. The line divides the DNN models between those producing moderate or better agreement on the testing set ($\kappa \geq 0.4$), and other where the agreement is low or non-significant ($\kappa < 0.4$). The plot shows that trained DNN models that performed very well in the validation set (i.e., values of val_acc close to 1.0) were also likely to produce the most accurate predictions on the testing set containing fingerprints of Abs never seen by the model. This analysis suggests that the $A_{val}$ metric may be used for selection of DNN models as well as a heuristic measure of prediction reliability.

We have outlined our approach and exemplar studies in the material above, but this methodology can be adopted and used by other in other areas of research. To this end we have made the computational tools publically available through a GitHub depository to researchers interested in pursuing this methodology in their own work. Key questions to be kept in mind is whether *1)* the property of interest of the Abs is directly determine by the amino acid composition and structural features of the Ab binding site region?, and whether *2)* there is sufficient structural information to associate a category or class to Abs sharing this specific property?

The amount of data required for efficient training of the DNNs depends on the complexity of the problem analyzed. For some applications, we used a single Ab to represent a class, but those classes were generally predicted with low accuracy. Based on our experience, better accuracies are obtained when the classes contain more than five Abs, as in the EBOV and HIV tests. Predictive accuracy can also be improved through the image augmentation procedure included in Keras.

What does this leave us? The approach presented in this work has been successfully applied for detecting Ab-epitope specificity and forms the foundation for investigating how to scale up the methodology to analyze larger number of sequences from B-cell repertoires. We also plan to expand the number of color schemes used for the fingerprints by incorporating other common motifs of molecular recognition observed in Ab-protein complexes such as cation-π interactions and the noncovalent attractive force between two aromatic rings, i.e., pi stacking.

## Summary

We designed a methodology and study to assess if the shape of the Ab binding sites together with the spatial distribution of key residues forming these sites could serve to discriminate Abs sharing a common characteristic, such as a preference for a given epitope from a larger pool of Abs. Our working hypothesis was that Abs targeting the same (or overlapping) epitope can only use a finite number of (residue) motifs on their Ab binding site regions to bind with high affinity to key elements on the antigen. A collection of Abs that bind to a particular epitope should thus provide a good sampling of the available motifs. The projection of residues generating these motifs onto a plane at the Ab-antigen binding interface defines a series of patterns that are captured in the Ab fingerprints. The DNN models–already pre-trained for image recognition–can then learn these patterns and recognize similar patterns in other Ab fingerprints.

Our work demonstrated that the reduced representations of the Ab binding site region could be used to train DNN models with predictive capacity that could sort and classify Abs based on their fingerprints. The trained DNNs were able to correctly infer the majority of the family lineages of Abs not included in the training sets. They also performed quite well as classifiers for detection of epitope specificity of Abs that were not included in the model construction *per se*. Based on these results, we consider that the application of the current methodology for detection of Ab binding to other antigenic determinants looks promising. Similarly, the modeling framework could also be used for one-class classification to separate and identify specific Abs from a much larger pool of previously unseen Abs, potentially paving the way for

deploying this technique in high-throughput sequencing of B cell repertoires as a tool to broaden the sequence diversity of lead antibodies for therapeutic usage.

Importantly, we showed that training of predictive DNNs could be accomplished by using the reduced information from fingerprints derived from an ensemble of Ab conformations generated with homology modeling techniques, and without explicit structural information of the antigen-Ab complexes themselves.

## Materials & methods

### Set of anti-EBOV Abs

Bornholdt *et al.* [23] cloned an extensive panel of mAbs targeting GP from peripheral B cells of a convalescent donor who survived the 2014 EBOV outbreak in Zaire [9,13]. The authors deposited the sequences of heavy- and light-chains of 349 mAbs in GenBank (accession numbers listed in S7 Table of their publication).

We used these Ab paired sequences together with the program BRILIA [24] to carry out a lineage assignment of the Abs. We compared our results with those reported by Bornholdt *et al.* [23] and found only few differences in the family assignment of Abs. We selected from these lineage families to train DNN models for Ab classification. S1 Table displays a list of the Abs included in ten of the most populated family lineages that we used for DNN classification, while S2 Table provides the lineage information and residue composition of the CDR regions. In addition, S2 Fig shows typical fingerprints associated with the Abs in the ten family lineages.

Multiple studies [23,41–43,52–58] have provided a detailed map of the epitopes in EBOV GP targeted by Abs. They showed that neutralizing mAbs preferentially target the GP1 head, fusion loop, base, and the α-helical heptad repeat 2 in the GP2 "stalk" (HR2) regions presented after enzymatic cleavage of GP. We used such data together with structural information and sequence analysis to construct three sets of Abs based on their epitope preferences (S3 Table).

We generated 17,643 3-D models with their respective fingerprints for 308 anti-EBOV Abs from the 349-antibody set reported by Bornholdt *et al.* [23] (see below for *Generation of 3-D models of antibodies*). We trained DNN models using fingerprints of selected Abs targeting two and three different epitopes on the EBOV GP trimer. Then, we tested these optimized models for detection of the binding preferences of other Abs, not included in the sets used for training and validation, which recognize the same epitopes. The selection of EBOV GP epitopes was carried out using data obtained from Fig 5 and S5 Table from Bornholdt *et al.* [23] publication.

We provide additional information on the set of anti-EBOV Abs used in this work in the S1 Text section: *Description of the set of anti-EBOV Abs*.

### Set of anti-HIV antibodies

HIV is a rapidly evolving virus that exists in many different viral strains for which no approved vaccine exists. Fortunately, a series of effective antiretroviral therapies developed during the last decade allows people carrying the disease to lead a healthy life [59]. It has been found that certain individuals infected with HIV can develop special type of antibodies, referred to as broadly neutralizing antibodies, capable of neutralizing a broad range of strains [60]. *CATNAP* (Compile, Analyze and Tally NAb Panels) is a web server [27] that provides access to an HIV *database with information* on neutralizing antibody sequences and potencies (e.g., $IC_{50}$ and $IC_{80}$), in conjunction with viral sequences for different strains. We used CATNAP, and the Protein Data Bank (PDB) [28] to retrieve sequences, binding data, and structural complexes pertaining to anti-HIV-1 Abs with gp120-gp41. We used PyMol [61] to align a large set of

structural complexes found in PDB using the gp120 molecule as target for the alignment. Based on this exercise, we identified the specific epitopes for the Abs and grouped them accordingly. We found that multiple Abs targeted two main binding regions in gp120-gp41 trimer, as shown in Fig 7 (Sites-HIV-1-trimer). The Abs targeting these two most populated binding sites were selected for following DNN analyses. S5 Table presents a list of 72 anti-HIV Abs with a description of relevant features that include light- and heavy-chain sequences, binding site and neutralization capacity, among others.

We generated 7,310 3-D models and fingerprints for 54 anti-HIV-1 Abs collected from PDB and CATNAP databases as described next.

## Generation of 3-D models of Abs

We used the PyRosetta and RosettaAntibody to generate 3-D models of the Abs described in this work [62–64]. Both, PyRosetta and RosettaAntibody are computational methods based on homology modeling techniques that uses the heavy- and light-chain sequences to predict the 3-D structure of antibodies. The programs contain custom databases constructed from high-quality structures in the PDB from which it automatically select optimal templates by requiring maximum sequence similarity with the chain sequences of the target Ab. To compute sequence similarity, a BLAST-based protocol is used [65].

In analogy with NMR structure determinations where the experimental structure is defined by a conformational ensemble, we used multiple 3-D models of the same Ab in an attempt to capture primarily the inherent flexibility of the CDRs that are the main determinants of the Ab binding site geometry. As an example, we show in Fig 13A the superposition of a single model for seven EBOV Abs that to recognize the stalk region of EBOV GP using a simplified representation. These models highlight charged residues located in the CDRs and show that equivalent residues in other Abs can occupy different positions in the 3-D space due to changes in the CDRs or the orientation of the side chains. Consequently, the fingerprints associated with these Abs may show very different patterns. By using multiple models for a given Ab, as shown in Fig 13B and 13C, we can generate multiple fingerprints with variations in their

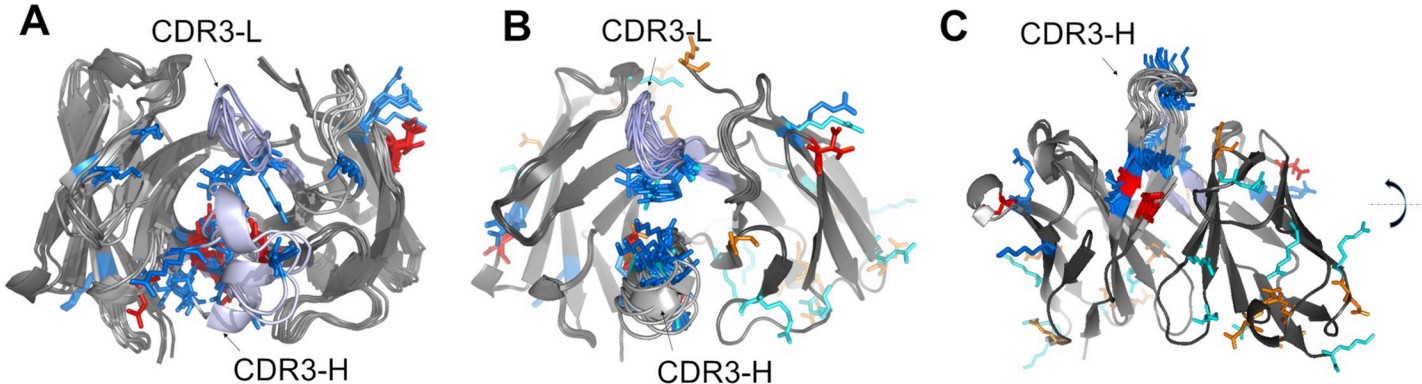

**Fig 13. Using multiple Ab models to account for the CDRs flexibility and variations of side-chain orientation.** (A) Superposition of 3-D models of seven EBOV Abs (ADI-15974, ADI-15756, ADI-15758, ADI-15999, ADI-15820, ADI-15848, ADI-16061) that target the stalk region of EBOV GP. For simplicity, the Abs are represented using grey ribbon models with positively- and negatively-charged residues associated with the CDRs shown with a 'stick' representation in blue and red, respectively. The light gray fragments of the ribbon models highlighted the positions of the light-chain CDR3s (CDR3-L), and heavy-chain CDR3s (CDR3-H). (B) Superposition of ten 3-D models of Ab ADI-15974 shown in the same orientation as those in (A) and using the same color scheme. Variations in the PDB templates used by Rosetta Antibody for 3-D models generation can lead to differences in the CDRs, and variations in the fingerprint patterns. In addition, for one of the models, we display the remaining positively- and negatively-charged residues of the Ab using cyan and orange colors, respectively. Note that projections of the latter set of residues may also contribute to the fingerprint patterns. (C) Same models as in panel B viewed using a 90° rotation around the horizontal axis.

patterns, with some of them resembling closer those present in the models of other Abs of the group. By training DNNs on the multiple patterns of each Ab, we expect that the trained DNN models will be able to extract consensus features that help identifying Abs with similar characteristics.

## Generation of fingerprints

For each Ab structural model, we built a two-dimensional map or *fingerprint* of the Ab binding site region. To generate the fingerprints, we colored amino acid residues according to a specified property such as residue charges, or using a simplified coloring scheme that groups the 20 amino acid residues in eleven types based on charge, hydrophobicity, hydrophilicity, aromaticity and other properties. In the former case, the color scheme based on the electrostatic charge of residues is positively charged residues, negatively charged residues, and non-charged residues are colored blue, red, and white, respectively. For the latter case, the color coding adopted is shown in Fig 1. It is worth noting that the amount of information contained in the sequence and the structural model of an Ab is substantially reduced in the fingerprints. There is no residue-connectivity information and, in particular, for fingerprints colored by charge, sequence and structural information of the Ab is reduced to a two-dimensional map of the Ab binding site based on a 3-color representation of the residues.

We wrote an implementation script that relies on a series of PyMol functions to produce fingerprints and associated image files. Briefly, we produced a *3-D template* of a generic Ab Fab using the backbone atoms from a structure from PDB. Separately, we generated a square *grid* that was positioned on top of the binding site region on the Ab template using the program PYMOL [61]. Every Ab structural model is aligned with the 3-D template and atoms of any residues in the Ab lying within a distance of 20 Å from the grid are projected onto the grid surface and colored according to the specified color scheme. To account in part for the shape of the Ab binding site region, we use a PYMOL depth model based on color attenuation for atom that are far from the grid, toward the Fab center. Brighter colors are used for atoms located closer to the grid or those that are part of protrusions traversing the grid toward the antigen.

## Deep neural network analysis

Deep learning is a class of machine learning procedure capable of decomposing raw inputs into multiple levels of representations with increasing levels of abstraction that are necessary for detection or classification. These higher layers of representation amplify particular features of the input that are useful for discrimination, while suppressing others that have no relevance during a classification task.

A Convolutional Neural Network (CNN) is a variant of a Deep Neural Network (DNN) consisting of a series of diverse layers that including convolutional, pooling, normalization, and fully connected layers that perform two main operations: feature extraction and classification. CNN are used quite successfully in the field of computer vision. The first few layers a CNN are able to extract automatically from the input data low-level meaningful features or key descriptors, such as edges, contours, and textures. Deeper layers in a CNN, on the other hand, extract features with higher level of complexity, such as four legs of an animal, a wheel, etc. In our applications, we used Keras [21], a deep learning API written in Python that runs on top of the machine learning platform TensorFlow [36], together with other Python packages for image processing, high-level mathematical functions for matrix manipulation, and statistical analyses (e.g., Numpy [66], Scikit-learn [25], Pandas [67], and others).

## Training and testing DNN for antibody classification on fingerprints

Transfer learning is a machine learning method in which a neural network model for a new application is built upon an existing model previously developed for a different task. The usage of pre-trained models as a starting point of model development is very common in computer vision and natural language processing applications, where model training on related problems will require a large amount of computer and time resources. We used this popular approach to develop some of our DNN models for antibody classification using the ResNet-50 model [22] with ImageNet weights.

In addition, we made use of Keras API for image augmentation. The process of image augmentation is carried out by taking images from the existing training dataset and manipulating them to create new altered versions. This procedure is quite useful when a reduced number of images is available for DNN training and aids in limiting overfitting. Thus, to augment the training set, new images are generated from the original ones via random transformations. Effectively, the DNN model never sees the same image more than once. These additional images may contain changes such a variations in coloring and lighting that can help produce a more robust classifier. We collect all the Ab fingerprints in a large pool of images and label each of them using the Ab name and a unique number.

In this work, we use the term *training* to refer to the optimization of DNN weights during the process of learning an Ab classification from a set of fingerprints. We use the term *testing* to indicate a subsequent application of the final DNN model to predict the class of Abs from a second set of images identified as the test set. Except for the experiment using pairs of Abs, the test set contains fingerprints from Abs that were not used during training of the DNN model. In general, we construct three sets of images, the training set, the validation set, and testing set for each DNN model. Fig 3 provides a schematic diagram of a typical assignment of Abs and fingerprints to these sets.

We generated a large database of 3-D models and associated fingerprints for the anti-EBOV GP, and anti-HIV-1 Abs sets. To train and evaluate a new DNN model, we first build the training, validation and testing sets as follows:

We produce a 'class' file for each class of Ab that the DNN needs to learn. The 'class' file contains a list of all the Abs assigned to the specific class. The assignment of the fingerprints is carried out automatically using a script that reads the antibodies of a given class from the specified 'class' file, and split them into two fractions in a random manner, one fraction is used for training and validation, the second fraction is used for testing. Typically, we use ~80% of the Abs for training and validation and ~20% for testing. After the partition of Abs from a class is defined, the process of allocating the fingerprints is initiated. When an Ab is selected for testing, all related fingerprints (a maximum number can be specified by the user) are added to a common pool (subdirectory) in the test set. Otherwise, all fingerprints of the specific Ab (a maximum number can be also be specified by the user) is randomly split in two fractions (e.g., (80% / 20%) between training and validation sets. Fingerprints assigned to the training or validation sets are added to separated pools corresponding to the Ab class (i.e., added to labelled subdirectories associated with the particular class).

Training of a DNN model involves a series of iterative steps or *epochs* in which images from the training and validation sets are used. The core of this process involves minimization of a cost function, which measures the error between predicted and expected values and reduces it to a single number. Because the image datasets are generally very large, model optimization cannot be carried out at once. The data must be split into batches to be passed to the optimizer. A complete round of optimization, or epoch, ends when all batches of images are used and a new set of weights is produced. After an optimization round ends, the performance of the

model with the new weights is evaluated using the images from the validation set. The number of epochs, and the number of images that constitute a single batch, or batch size, are user-defined parameters. The number of epochs must be chosen properly to achieve converge of the optimization process.

To carry out the optimization task, we used *Adam*, an algorithm for first order gradient-based optimization of stochastic objective functions provided in the Keras API. We chose Keras *binary_crossentropy* as the loss function, *L*, when training a model for binary classification. The loss is evaluated as:

$$L(\mathbf{y}, \mathbf{p}) = -1/N \sum_{j=1}^{N} (y_j \cdot \log(p_j) + (1-y_j) \cdot \log(1-p_j)) \tag{1}$$

where $y_j$ is a binary indicator whose value is "1" if the *j* observation belong to the class designated as correct, and "0" otherwise, $p_j$ represents the predicted probability that the the *j* observation is correct, and N is the total number of predictions.

On the other hand, we use Keras *categorical_crossentropy* as the loss function for models involving multiclass classifications. In this case, the loss is computed as:

$$L(\mathbf{y}, \mathbf{p}) = -\sum_{j=1}^{M} \sum_{j=1}^{N_i} (y_{ij} \cdot \log(p_{ij})) \tag{2}$$

where $y_{ij}$ is a binary indicator whose value is "1" if the *j* observation belong to the class *i*, and "0" otherwise; $p_{ij}$ represents the predicted probability that the *j* observation belongs to class *i*, $N_i$ is the total number of predictions corresponding to class *i*, and *M* corresponds to the total number of classes.

To assess the quality of the DNN model during training, we used the validation accuracy ($A_{val}$), a metric that measures the accuracy of the model on the validation set. We carry out the final assessment of the DNN model using fingerprints from the test sets, and the accuracy of the model in this set is reported as $A_{test}$.

## One-class classification

One-class classification (OCC) methods attempt to recognize instances of a given class, i.e., the normal class, from a large pool of instances belonging either to the normal class or to a second generic class. The latter, denotes as the "anomalous" class, may include multiple other classes with the exception of the normal one. The training of these classifiers is carried out using only instances of the normal class, while instances of the anomalous class are encountered during the validation phase. OCC algorithms are expected to capture the density of the normal class and classifies examples on the extremes of the density function as anomaly or outliers. For our calculations, we used the Robust Convolutional Autoencoder (RCAE) [35], an unsupervised anomaly detection technique that separate normal from anomalous data using an autoencoder. The latter is an unsupervised neural network that compress the data into an encoded representation in an inner hidden layer, and learn how to reconstruct the original data back from its reduce representation, i.e., to decode it. The RCAE decomposes an input data X into two parts $X = L_D + S$, with $L_D$ being a latent representation of the hidden layer of the autoencoder, and *S* is a matrix that captures as noise those features of the outliers that are hard to reconstruct. Chalapathy *et al.* [35,39] proposed to carry out the decomposition of the input data through optimization of the following objective function:

$$\min_{\theta, S} \|L_D - D_\theta(E_\theta(L_D))\|_2 + \lambda.\|S^T\|_{2,1} \tag{3}$$

$$\text{s.t. } X - L_D - S = 0,$$

where $D_\theta$ and $E_\theta$ are generic functions with parameter $\theta$ that represent the decoder and encoder, respectively, and $\lambda$ is a tuning parameter.

In one of the studies presented in the Results Section, we applied the OCC methodology to detect new Abs from a family lineage not seeing during training, we trained our models using fingerprints from Abs belonging to a single-family lineage as the "normal" class, and used the fingerprints from the remaining nine families as outliers. Fig 14 shows a schematic diagram describing the construction of the fingerprint datasets used for training and evaluation of the DNN models for OCC.

**Antibody assignment.** Antibodies from a single lineage were considered the "normal" class while the rest of the Abs from other families were considered the "outliers" or "anomalous" class. The latter group of Abs was divided into two subsets at ~80/20 ratio.

**Fingerprints assignment.** All fingerprints from one Ab in the "normal" class were added to the testing set for a performance evaluation of the final DNN model, while fingerprints from the rest of the "normal" Abs were split between training and validation sets at ~67/33 ratio. Fingerprints from the largest subset (~80%) of "anomalous" Abs were added to the validation set as negative control. A fraction of fingerprints from the small subset of "anomalous" Abs was selected randomly for inclusion into the testing set. The number of fingerprints from "normal" and "anomalous" Abs in the testing set was approximately the same. We used a similar approach to assign the Abs and fingerprints in other studies using the OCC method.

For these experiments, we resized the fingerprint images of the Abs to 32x32x3 pixels to conform to the requirement of the RCAE algorithm. Optimization of the loss function was carried out using the first order gradient-based optimizer Adam. We set the number of Epochs, a parameter used to define the optimization cycles, to values ranging from 100 to 200.

## Statistical analysis of multi-class predictions

We used different types of metrics to evaluate the performance of our models. We compute the following quantities:

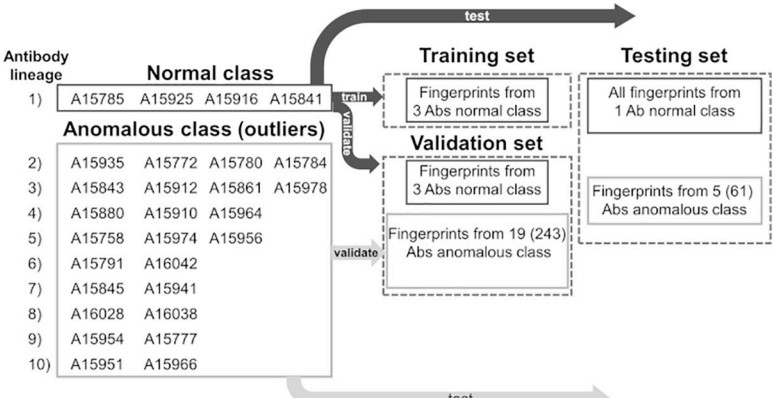

**Fig 14. Schematic diagram of the allocation of Ab fingerprints into training, validation, and testing sets for one-class classification.** See text for an explanation of antibody and fingerprints assignment. *Note*: the Abs labels have been simplified where "A" stands for "ADI-".

*Accuracy* as the ratio between the number of correct predictions and the total number of input samples;

$$\text{Accuracy} = \frac{\text{TP} + \text{TN}}{\text{TP} + \text{FP} + \text{FN} + \text{TN}} \tag{4}$$

*Precision* as the number of correct positive results divided by the number of positive results predicted by the classifier;

$$\text{Precision} = \frac{\text{TP}}{\text{TP} + \text{FP}} \tag{5}$$

*Recall (or sensitivity)* as the number of correct positive results divided by the number of all relevant samples;

$$\text{Recall} = \frac{\text{TP}}{\text{TP} + \text{FN}} \tag{6}$$

*F1-score* as the Harmonic Mean between *precision* and *recall*;

$$\text{F1 Score} = 2 \cdot \frac{(\text{Recall} \cdot \text{Precision})}{(\text{Recall} + \text{Precision})} \tag{7}$$

Values of the F1-score range between 0 and 1. It measures a test's accuracy, and indicates how precise and robust the classifier is.

## Confusion matrix for multi-class prediction

The confusion Matrix describes the complete performance of the model. Confusion Matrix forms the basis for the other types of metrics.

## Precision local, recall local & F1-score local

To produce these measures we first compute for a given class *j*, $\text{TP}^j$, the true positives for class *j*, as the sum of the TPs that each of the DNN model produced for all Abs in class *j*. Similarly, we obtain the false positives for class *j*, $\text{FP}^j$, and the false negatives for class *j*, $\text{FN}^j$. We compute precision local, recall local, and F1-score local by replacing TP, FP, and FN in equations [5], [6], and [7], respectively, with $\text{TP}^j$, $\text{FP}^j$, and $\text{FN}^j$, respectively.

## Micro averages

In a micro-average calculations we compute the sum of the individual true positives ($\text{TP}_{\text{tot}}$), false positives ($\text{FP}_{\text{tot}}$), and false negatives ($\text{FN}_{\text{tot}}$) produced by all the DNN models for all classes. Then, to obtain the micro-average precision, micro-average recall and micro-average F1-score, we compute precision, recall and F1-score by replacing TP, FP, and FN in Equations [5], [6], and [7], by $\text{TP}_{\text{tot}}$, $\text{FP}_{\text{tot}}$, and $\text{FN}_{\text{tot}}$, respectively.

## Macro averages

In the macro-average calculations, we compute the macro-average precision, macro average-recall and macro-average F1-score as the unweighted averages of precision local, recall local, and F1-score local, respectively, over all the classes.

## Weighted averages

*In contrast to a macro-average*, weighted-average precision, weighted-average recall and weighted-average F1-score are calculated as weighted averages of precision local¸ recall local, and F1-score local for each class, respectively, where the weight associated with a given class is computed as the number of true instances of such class divided the total number of instances for all the classes.

A macro average provides a measure of how the system performs overall across the datasets. On the other hand, a micro average is a useful evaluation of the performance when the size of the datasets varies. Weighted-averaging favors those classes with large number of instances.

## Generalized kappa coefficient

To assessment the quality of multiclass predictions, we use the generalized Kappa coefficient introduced by Gwet [68].

## Area under the curve

We use *Area Under the Curve* (AUC) for analysis of binary classification problems. The AUC measures the capability of the classifier to distinguish between classes.

We compute the *Area Under the Receiver Operating Characteristic curve (AUROC)* from prediction scores. AUROC is a representation of the True Positive Rate (TPR) versus False Positive Rate (FPR) at various classification threshold within the range [0,1]. The TPR, also known as recall or sensitivity, is calculated using Eq 6. The FPR is computed as;

$$\text{FPR} = \frac{\text{FP}}{\text{FP} + \text{TN}} \tag{8}$$

We computed these statistical measures using the Python Scikit-learn library for machine learning and statistical modeling [25].

## Understanding DNN predictions with LIME

The Local Interpretable Model-agnostic Explanations (LIME) is a novel technique that attempts to explain the predictions of any classifier or regressor by generating an interpretable model locally faithful around a given prediction [69]. In those cases involving image classification as the ones presented in this work, a classifier can represent an image as a tensor with three color channels per pixel. LIME can analyze the output from such a classifier to produce an interpretable representation as binary vector that describes the presence or absence of a contiguous patch of pixels with similar importance.

Let us consider the application of a classification model to a given instance x, and let f(x) be the probability that x belongs to a particular class produced by such model. LIME generates an explanation of such assignment as a new model g that belongs to a class G of potentially interpretable models, Formally, LIME's explanation is given by the following equation:

$$\xi(x) = \text{argmin } L(f, g, \pi_x) + \Omega(g), \tag{9}$$

where the loss function $L(f, g, \pi_x)$ represents a measure of how unreliable is g as an approximation of f; $\pi_x$ is a proximity measure between an instance z to x that define locality around x; and $\Omega(g)$ is a measure of complexity of the explanation g.

Expression 9 indicates that the explanation obtained by LIME corresponds to the argument that minimizes L by keeping $\Omega(g)$ low enough to be interpretable by humans.

For additional explanation of the methodology, the user is referred to the original document describing LIME [69].

## Supporting information

**S1 Text. Description of the set of anti-EBOV Abs.**
(DOCX)

**S1 Fig. Proposed approach to assess high-throughput B cell immune-sequencing data with Artificial Intelligence methods.** (A) Traditional image recognition problem in which a DNN model is used to identify or classify objects, e.g., cats from dogs, different breeds, etc. Images were obtained from Wikimedia Commons (see S9 Table for a list of credits and reproduction license agreements). (B) For the goal of developing AI techniques capable of Ab characterization based on B cell sequence, we need to develop a number of related methodological capabilities that included conversion of sequences into image representations to enable identification and classification by means of DNNs.
(DOCX)

**S2 Fig. Fingerprints of twenty-eight anti EBOV Abs associated with ten family lineages.** Family lineages are labeled L1 to L10. Fingerprints corresponding to all Ab members of a family are displayed in a column. The order in which Abs are listed in a family is arbitrary. Note: the Abs labels have been simplified where "A" stands for "ADI-". The ID numbers correspond to the Abs described in reference [23].
(DOCX)

**S1 Table. Identification numbers of anti EBOV antibodies from ten family from the most populated lineages.** The ID numbers correspond to the Abs described in reference [23].
(DOCX)

**S2 Table. Sequence analysis of anti EBOV antibodies from ten family lineages.** Sequence information summary from BRILIA [24] for 28 anti-EBOV antibodies used to train and test DNN models for family lineage detection. The charged residues found within the CDRs are highlighted in the respective columns listing the sequences.
(DOCX)

**S3 Table. Sets of anti EBOV antibodies used for DNN training and testing for epitope recognition.** Abs in $Set_1$ and $Set_2$ bind to the GP1 base. Abs in $Set_3$ target the α-helical heptad repeat 2 in the GP2 "stalk" (HR2) region. The ID numbers correspond to the Abs described in reference [23].
(DOCX)

**S4 Table. Detection of lineage family.** Summary of 40 DNN models used for classification of 28 antibodies belonging to ten family lineages using fingerprints colored using the reduced amino acid alphabet color-coding.
(DOCX)

**S5 Table. Properties of the HIV-1 antibodies used in our study.** This Table is provided as an EXCEL file (S5_Table.xlsx), and includes Ab names, sequences, binding site specificities, and PDB codes for the experimental structures of the HIV-1 Abs and their complexes.
(XLSX)

**S6 Table. Detection of Abs from diverse clonotypes using the RCAE method.** Sequence annotations of Abs from the normal class included in the training set. The sequence analysis

was carried out using BRILIA [24].
(DOCX)

**S7 Table. Detection of Abs from the same EBOV competition group using the RCAE method.** Statistical summary of the 10 top DNN models trained on anti EBOV Abs from a single-family lineage, and used to distinguish other Abs from the same competition group.
(DOCX)

**S8 Table. Sequence comparisons of the Abs detected by the RCAE method with Abs known to bind to the base of EBOV GP$_{1,2}$.**
(DOCX)

**S9 Table. List of cat and dog images used to produce S1 Fig.** Images were obtained from Wikimedia Commons (commons.wikimedia.org) & Creative Commons (creativecommons.org)
(DOCX)

## Acknowledgments

The authors thank Mr. Michael Madore for technical assistance on software implementation. The opinions and assertions contained herein are the private views of the author(s) and are not to be construed as an official position, policy or decision of Department of the Defense or the Henry M. Jackson Foundation for Advancement of Military Medicine, Inc. unless so designated by other documentation. This paper has been approved for public release with unlimited distribution.

## Author Contributions

**Conceptualization:** Daniel R. Ripoll, Sidhartha Chaudhury, Anders Wallqvist.

**Data curation:** Daniel R. Ripoll.

**Formal analysis:** Daniel R. Ripoll, Sidhartha Chaudhury, Anders Wallqvist.

**Funding acquisition:** Sidhartha Chaudhury, Anders Wallqvist.

**Investigation:** Daniel R. Ripoll, Sidhartha Chaudhury, Anders Wallqvist.

**Methodology:** Daniel R. Ripoll, Sidhartha Chaudhury, Anders Wallqvist.

**Project administration:** Sidhartha Chaudhury, Anders Wallqvist.

**Resources:** Anders Wallqvist.

**Software:** Daniel R. Ripoll.

**Supervision:** Sidhartha Chaudhury, Anders Wallqvist.

**Validation:** Daniel R. Ripoll, Anders Wallqvist.

**Visualization:** Daniel R. Ripoll, Anders Wallqvist.

**Writing – original draft:** Daniel R. Ripoll, Sidhartha Chaudhury, Anders Wallqvist.

**Writing – review & editing:** Daniel R. Ripoll, Sidhartha Chaudhury, Anders Wallqvist.

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
