## [Decision Letter · Decision Letter 0]

2 Nov 2020

Dear Dr. Wallqvist,

Thank you very much for submitting your manuscript "Using Paratope Features to Train Deep Neural Networks for Antibody Classification" for consideration at PLOS Computational Biology.

As with all papers reviewed by the journal, your manuscript was reviewed by members of the editorial board and by several independent reviewers. In light of the reviews (below this email), we would like to invite the resubmission of a significantly-revised version that takes into account the reviewers' comments.

The reviewers have raised significant concerns with the paper in particular whether the method offers any real advantages over much faster sequence based methods which must be fully addressed in any revised submission.

We cannot make any decision about publication until we have seen the revised manuscript and your response to the reviewers' comments. Your revised manuscript is also likely to be sent to reviewers for further evaluation.

Sincerely,

Charlotte M Deane

Associate Editor

PLOS Computational Biology

Nir Ben-Tal

Deputy Editor

PLOS Computational Biology

The reviewers have raised significant concerns with the paper in particular whether the method offers any real advantages over much faster sequence based methods which must be fully addressed in any revised submission.

Reviewer's Responses to Questions

**Comments to the Authors:**

Reviewer #1: Ripoll et al. present a novel machine learning method trained on in vitro-confirmed epitope binders to classify antibody binding sites by their likely epitope engagement. Their goal is that this method could be applied to high-throughput sequencing (HTS) datasets to more reliably identify the subset of antibodies that bind to a particular antigen epitope. The current state-of-the-art, given a set of known binders to an epitope, is to search through HTS datasets for antibodies of very close genetic relatedness (same closest V/J genes and high CDRH3 sequence identity, `clonotyping’) and to shortlist these as likely to be functionally equivalent. Ripoll et al.’s proposed method is compelling, as it goes beyond a purely sequence-based approach by taking as input flattened 2D representations of 3D-models of each antibody binding site; such explicit topological consideration could therefore identify genetically-similar antibodies that have too different a topology to bind to the same epitope, reducing false positives. Their proposed framework also has the potential, if trained on a sufficiently diverse set of clonally distinct antibodies that can bind to the same binding site, to capture the crucial topological and chemical features that govern epitope complementarity, and thus to be able to cluster more diverse antibodies than clonotyping with likely functional commonality. Proof of either of these two advances would represent a significant step forward for the field.

As the manuscript is currently constructed, it is impossible to say whether their DNN architecture is sufficiently sensitive to detect the former category (i.e. discern between sequence-similar antibodies that can/cannot engage the same epitope). The only antibodies used in training are ones that are proven to bind to one of multiple epitopes on an antigen's surface, and each epitope’s binders appear to be highly genetically distinct (judging by the EBOV binders to 3 epitopes, sourced from GenBank via. Bornholdt et al.). The algorithm would need to see sequence-similar negative examples, i.e. antibodies that are close in sequence to known binders but whose binding capabilities are destroyed by particular mutations/the use of signficantly different CDR3s.

It is possible that the method can detect clonally distinct antibodies that can bind to the same epitope, but it is not clear from how the manuscript is currently presented and may depend on the tested epitope. EBOV epitope 1, for example, is so homogenous in its binding examples that the DNN could learn trivial classification features (11/12 binders belong to the IGHV4-34/IGHJ4 germline, of which 9 have at least one partner from the same VH clonotype [see example]; and no other epitope has an antibody from the IGHV4-34 germline that can bind to it). Atomic patterns of the IGHV4-34 CDRH1/CDRH2 alone could therefore dominate the classification parameters and may in practise lead to any IGHV4-34-derived antibody that is fed into the model being predicted as an Epitope 1 binder (highly unlikely to be meaningful when applied to HTS). Relatedly, it seems strange that so many tables (3, 9, 10) describe the results of applying the DNN to detect antibody lineages. As the authors say, this is already trivial by sequence, and in my view the exciting feature of this methodology is that it has the potential to see beyond lineage to tell researchers orthogonal information about which chemical/topological features are necessary to bind to a particular epitope.

EXAMPLE

ADI-15843: IGHV4-34 + IGHJ4 + ARAWLRSRGYPSFDY + IGKV2-28 + IGKJ5 + MQALQTLT

ADI-15912: IGHV4-34 + IGHJ4 + ARAWLRSRGYPSFDY + IGKV2-28 + IGKJ5 + MQALQTLT

These two antibodies to the same EBOV epitope differ by just 3 amino acid residues across the entire Fv. Fingerprints of an identical VJH3 + VJL3 lineage are therefore currently allowed to co-exist in the training and test set.

I would like to see:

(A) evidence (e.g. through LIME) that the most important features used to distinguish binders to every epitope are spread in a meaningful way throughout the CDRs.

(B) evidence that good model performance is not simply dependent on testing on an example with a same-VH clonotype binder in the training set (test accuracy seems to vary widely depending on the training/testing split). This could be provided by not only blinding the model at training-time to fingerprints from the tested antibody, but also to those of same-epitope binding antibodies from the same clonotype/lineage. LIME could also be used here to look at how the important features change based on this blinding procedure. If the model cannot perform well without knowledge of a same-lineage binder, it would offer little more practical value than clonotyping (if I already knew an antibody from that lineage bound that epitope, then clonotype “fishing” can currently highlight antibodies from the same lineage as worthy of investigation).

(C) An OCC experiment trained with the "normal class" of antibodies that can bind the same epitope (again blinded to antibodies from the same lineage as the tested antibody), rather than antibodies deriving from the same lineage family, to see whether the model can provide more insight than would be provided by sequence-based lineage clustering alone.

Other general points:

- The method is notable in the fact that it could capture the influence of the light chain CDRs on binding specificity. As most deep antibody repertoire sequencing datasets are currently heavy chain only, it would be interesting to see the effect on predictive performance if the light-chain component of the fingerprints is removed.

- The authors should explain in detail the origin of the stochasticity generated by modelling the same antibody multiple times and explain why it is a good proxy for binding site flexibility. I could not find this described in the Methods section.

- Certain methodological changes (e.g. charge colour scheme/reduced amino acid alphabet and image enhancement/no image enhancement) should be discussed - why the do the authors believe the chosen representations can yield such different accuracies?

- The authors state “comparison of the 3-D structure of every Ab in a repertoire is unfeasible”. They should clarify that this refers to experimental structure determination, as published software already exists to computationally perform full-repertoire structural comparisons (e.g. SAAB+, 10.1371/journal.pcbi.1007636)

Reviewer #2: Using Paratope Features to Train Deep Neural Networks for Antibody

Classification

The authors use a very novel and clever approach to attempt to discover patterns in antibody sequence/structural motifs that allow for prediction of specificity to antigen.

While it is very clever to use 3D structural predictions of paratope combined with biophysical features to then create a 2D-image of paratope, there are questions as to whether this level of elaborative methods is necessary compared to more conventional sequence based tools.

A major question is to what extent is the 3D-structural model for Rosetta adding important information? If it is to simply model and thus define paratopes residues, then I am not convinced that it is a requisite step. Nearly all antibody paratopes are driven by CDRs, especially CDRL3 and CDRH3, and thus simply extracting CDR sequences and using them as the paratope would be a simple solution. CDRs could even be weighted as has been previously done for TCRs (see Dash, Nature, 2017).

By using a 3D model that has no certainty in accurately modeling the dynamic conformational loops of antibody CDRs only to define a slightly more minimal paratope residues does not seem like an effective strategy. Most notably it dramatically increases the time and computational power required to perform this on large sequencing datasets. If the authors are convinced about this approach, then they should directly compare the results of their classification scheme using 3D-structural models directly with simple CDR sequence based paratope definitions.

“We note that the detection of family lineages is easily achieved with computational tools based on sequence analysis. Our objective, however, was to determine the ability of DNNs to learn to associate members of the lineage family using similarities in the image patterns based on the arrangement of color on the fingerprints.” I appreciate stating clearly that sequence analysis tools are already sufficient to cluster antibodies into family lineages and thus they simply wanted to benchmark their DNNs. But based on the data in Table 2, they achieve a global accuracy of their DNNs of 0.62. This seems extremely low when compared to simple sequence based lineage clustering. With a performance at this level the authors have to justify what value their approach really adds for lineage classification. For example, if they could classify sequences to a lineages with DNNs that fall outside of common sequence similarly metrics (e.g., 80% similarity of CDRH3) that would add value and convince me of the potential of their approach. They should thus attempt to look at this aspect.

Similar to my point above, the authors perform a classification analysis on Ebola and HIV antibodies, trying to accurately predict epitope groups. In many cases antibody sequences that bind the same epitope may have very similar germline or CDR sequences. Thus the authors should compare whether very simple machine learning classification models based on antibody sequences (germline or CDRs) could similarly predict epitope specificity. For example the aforementioned TCRdist (by Dash, Nature, 2017) was able to predict specificity of TCRs using a weighted distance-based clustering of CDR sequences followed by a k-nearest neighbor model to accurately predict TCRs specificity. Something similar could in principle work for antibodies as well and serve as an important benchmark to compare their approach.

Reviewer #3: Summary

In this work, Ripoll and colleagues developed a paratope-based DL-classification approach. They use this approach to classify Abs belonging to, among others, different lineages. While this work is interesting, the authors fail to acknowledge prior literature and do not compare their approach to already existing approaches. The relevance of their results is thus questionable. The manuscript has a lot of redundant text. It could benefit from strict editing. Major and minor issues are written below.

Major

Prior relevant literature seems to be unknown to the authors.

Just a few examples because there are too many:

“Comparison of the 3-D structure of every Ab in a repertoire is unfeasible.” → please see https://www.frontiersin.org/articles/10.3389/fimmu.2018.01698/full and follow-up papers.

“The data derived from immunological studies typically represent the result of ongoing stochastic and multifactorial processes that is often difficult to decipher. Artificial Intelligence (AI) methods are potentially well suited to address these types of problems. Thus, machine learning approaches have already been used for predicting peptide presentation by T cells (5-7), affinity of peptide binding to Major Histocompatibility Complex molecules (8), and binding affinity of neutralizing Abs (9). Deep learning techniques have also been used for de novo prediction of protein structures (10, 11).” → I suggest the readers have a look at reviews such as: https://academic.oup.com/bib/article/21/5/1549/5581643 or https://www.sciencedirect.com/science/article/pii/S2452310020300524 instead of erratically citing barely relevant papers.

“The epitope binding site on the antigen itself is not used as we typically do not have this information.” → If you don’t have the epitope, you, by definition, also don’t have the paratope. How was the paratope delineated for the Ab sequences studies?

Can you provide an overview table with the number of Abs in Train, Val, and Test datasets? This paper contains a lot of tables. Can you maybe summarize them all together in one figure using boxplots/barcharts. In such a figure, you could then also visualize the research of a given ML experiment, which would make this manuscript much clearer. It's overall very confusing to read.

Can the authors compare their method to already existing approaches such as: https://academic.oup.com/bioinformatics/article/34/17/2944/4972995 or with simpler baseline approaches as Logistic regression? Or at least discuss why they did not compare with prior literature?

Can you provide sequence similarity quantification of the datasets to be classified to clarify the a priori difficulty of the classification challenge? In other words, can you quantitatively motivate why machine learning/deep learning is necessary for your research question?

“1) Can we differentiate Abs based solely on paratope features? 2) Can we identify sets of Abs produced by B cells that originated from a common ancestor, i.e., the family lineage?”

→ ad 1) It’s unclear what this question means? Differentiate based on what?

→ ad 2) Why not use phylogenetic analysis?

Can you discuss to what extent your approach can be applied to datasets where extensive lineage information is not available? Is your approach dependent on prior 3D-information?

Figure 1: this figure is trivial. This is not a review. Please add some information that we don’t know already. You are not the only one thinking of image classification: https://www.biorxiv.org/content/10.1101/2019.12.18.880146v2

Minor

The human immune system is capable of producing on the order of 10^20 antibodies (Abs) in response to a viral infection. → citation for 10^20?

**Have all data underlying the figures and results presented in the manuscript been provided?**

Reviewer #1: Yes

Reviewer #2: Yes

Reviewer #3: Yes

PLOS authors have the option to publish the peer review history of their article (what does this mean?). If published, this will include your full peer review and any attached files.

Reviewer #1: No

Reviewer #2: No

Reviewer #3: No
---

## [Decision Letter · Decision Letter 1]

28 Feb 2021

Dear Dr. Wallqvist,

Thank you very much for submitting your manuscript "Using Paratope Features to Train Deep Neural Networks for Antibody Classification" for consideration at PLOS Computational Biology. As with all papers reviewed by the journal, your manuscript was reviewed by members of the editorial board and by several independent reviewers. The reviewers appreciated the attention to an important topic. Based on the reviews, we are likely to accept this manuscript for publication, providing that you modify the manuscript according to the review recommendations.

Sincerely,

Charlotte M Deane

Associate Editor

PLOS Computational Biology

Nir Ben-Tal

Deputy Editor

PLOS Computational Biology

[LINK]

Reviewer's Responses to Questions

**Comments to the Authors:**

Reviewer #1: I thank the authors for making significant improvements to the design of their investigation, in particular removing any sequence redundancy and adding a crucial OCC experiment where the normal class compromises different-lineage antibodies that engage the same binding site. The results act as a proof-of-concept that their methodology (at least in 4-6% of models) can perform the difficult task of "seeing past" lineage to bin together less related antibodies able to compete for the same epitope.

While this tool shows promise, its typical performance (and in particular its significantly lower performance than sequence-based methods at binning same-lineage same-epitope binders together) is likely to limit its immediate wider adoption by the community. Nonetheless, this drop in performance is perhaps unsurprising given the need to relax the similarity threshold to capture more distantly-related antibodies that bind the same epitope with different binding modes. This paper represents a thorough and useful initial benchmark of the performance of a 2D image-based DNN for antibody epitope binning. I would be interested to see just how much the performance improves when trained on binders to more intensely studied epitopes, such as the large number of antibodies now shown to compete for the ACE-2 binding site of the SARS-CoV-2 spike receptor binding domain. Overall, I could see this published in PLoS Computational Biology as a valuable initial yardstick against which to measure future structure-aware antibody function classification algorithms.

As an aside, I agree with reviewer 3 that using the term “paratope features", in the title and throughout the manuscript, is too strong as you are neither predicting nor evaluating the paratope ahead of feature selection. From your graphic, it appears that you are equating the term 'paratope' with the CDRs. In addition, the use of the term “classification” in the title, while accurate, does not communicate the ultimate intended application of the method. Since you settled on an APN residue alphabet, I would suggest the following title: “Using the Spatial Distribution of Complementarity-Determining Region Charges to Train Deep Neural Networks for Antibody Epitope Binning”, or some variation thereof.

Reviewer #3: All of my comments have been addressed.

**Have all data underlying the figures and results presented in the manuscript been provided?**

Reviewer #1: Yes

Reviewer #3: Yes

PLOS authors have the option to publish the peer review history of their article (what does this mean?). If published, this will include your full peer review and any attached files.

Reviewer #1: No

Reviewer #3: No

Figure Files:

Data Requirements:

Reproducibility:

References:

---

## [Editor Report · Decision Letter 2]

10 Mar 2021

Dear Dr. Wallqvist,

We are pleased to inform you that your manuscript 'Using the Antibody-Antigen Binding Interface to Train Image-Based Deep Neural Networks for Antibody-Epitope Classification' has been provisionally accepted for publication in PLOS Computational Biology.

Best regards,

Charlotte M Deane

Associate Editor

PLOS Computational Biology

Nir Ben-Tal

Deputy Editor

PLOS Computational Biology

---

## [Editor Report · Acceptance letter]

24 Mar 2021

PCOMPBIOL-D-20-01791R2 

Using the Antibody-Antigen Binding Interface to Train Image-Based Deep Neural Networks for Antibody-Epitope Classification

Dear Dr Wallqvist,

I am pleased to inform you that your manuscript has been formally accepted for publication in PLOS Computational Biology. Your manuscript is now with our production department and you will be notified of the publication date in due course.

With kind regards,

Katalin Szabo
